# High-fat diet-induced upregulation of exosomal phosphatidylcholine contributes to insulin resistance

Anil Kumar [1], Kumaran Sundaram[1], Jingyao Mu[1], Gerald W. Dryden[1,2], Mukesh K. Sriwastva[1], Chao Lei[1], Lifeng Zhang[1], Xiaolan Qiu[1], Fangyi Xu[1], Jun Yan[1], Xiang Zhang[3], Juw Won Park [4,5], Michael L. Merchant[6], Henry C. L. Bohler[7], Baomei Wang[8], Shuangqin Zhang[9], Chao Qin[10], Ziying Xu[10], Xianlin Han [10], Craig J. McClain[2], Yun Teng[1,12 ✉] & Huang-Ge Zhang [1,11,12 ✉]

High-fat diet (HFD) decreases insulin sensitivity. How high-fat diet causes insulin resistance is largely unknown. Here, we show that lean mice become insulin resistant after being administered exosomes isolated from the feces of obese mice fed a HFD or from patients with type II diabetes. HFD altered the lipid composition of exosomes from predominantly phosphatidylethanolamine (PE) in exosomes from lean animals (L-Exo) to phosphatidylcholine (PC) in exosomes from obese animals (H-Exo). Mechanistically, we show that intestinal H-Exo is taken up by macrophages and hepatocytes, leading to inhibition of the insulin signaling pathway. Moreover, exosome-derived PC binds to and activates AhR, leading to inhibition of the expression of genes essential for activation of the insulin signaling pathway, including IRS-2, and its downstream genes PI3K and Akt. Together, our results reveal HFD-induced exosomes as potential contributors to the development of insulin resistance. Intestinal exosomes thus have potential as broad therapeutic targets.

[1] James Graham Brown Cancer Center, Department of Microbiology & Immunology, University of Louisville, Louisville, KY 40202, USA. [2] Department of Medicine, University of Louisville, Louisville, KY 40202, USA. [3] Department of Pharmacology and Toxicology, University of Louisville, Louisville, KY 40202, USA. [4] Department of Computer Engineering and Computer Science, University of Louisville, Louisville, KY 40202, USA. [5] KBRIN Bioinformatics Core, University of Louisville, Louisville, KY 40202, USA. [6] Kidney Disease Program and Clinical Proteomics Center, University of Louisville, Louisville, KY, USA. [7] Department of Reproductive Endocrinology and Infertility, University of Louisville, Louisville KY40202, USA. [8] Department of Dermatology, University of Pennsylvania, Philadelphia 19104, USA. [9] Peeples Cancer Institute, 215 Memorial Drive, Dalton, GA 30720, USA. [10] Barshop Institute for Longevity and Aging Studies, University of Texas Health Science Center at San Antonio, San Antonio, TX 78229, USA. [11] Robley Rex Veterans Affairs Medical Center, Louisville, KY 40206, USA. [12] These authors jointly supervised this work: Yun Teng, Huang-Ge Zhang. ✉email: yun.teng@louisville.edu; h0zhan17@louisville.edu

The global escalation of obesity and diabetes poses a great health challenge. Insulin resistance is a hallmark of type 2 diabetes (T2D) and is associated with metabolic disorders, yet the precise interplay between the molecular pathways that underlie them is not fully understood. The accumulation of bioactive lipids in nonadipose tissues has been proposed to promote impaired insulin sensitivity[1–3]. Abnormally high cellular phosphatidylcholine (PC) lipid influences energy metabolism and is linked to insulin resistance[4–8].

High-fat diets (HFD) represent a public health concern as they can predispose individuals to obesity and diabetes[1,9,10] and promote overproduction of PC and insulin resistance[11]. From a physiological point of view, one of the most important links between the high-fat diet and insulin resistance is the gut–liver axis[12] and the factors released from intestinal and liver metabolites, which mediate a bidirectional communication between the intestines and the liver. The gut–liver axis is impaired following consumption of HFD.

Recent data indicate that extracellular vesicles (EVs) play a role in obesity and the development of its metabolic complications by serving as a mode of intercellular communication among adipose tissue, liver, skeletal muscle, and immune cells. Exosomes may play a role in the regulation of peripheral insulin sensitivity, a major component of the pathogenesis of T2D mellitus[13]. Exosomes appear to regulate insulin sensitivity through at least two different mechanisms, i.e., by modulating inflammation or by direct interaction with insulin-responsive organs. The latter may occur directly or indirectly by affecting insulin signaling pathways[14] through interaction with major insulin signaling molecules, such as the phosphatidylinositol-3-kinase (PI3K)/Akt signaling pathway. A higher level of plasma EVs is found in patients with diabetes compared to controls, and a consistent trend is also reported in mice with diabetes that exhibited a higher number of plasma exosomes compared to control mice[15–17]. However, defining the role of exosomes and their potential role in liver/gut axis communication will require a better understanding of their composition, in particular, whether diet alters the content of intestinal-derived exosomes and hence their biological functions in terms of insulin response has not been studied.

One protein of interest in regulating the insulin response is the aryl hydrocarbon receptor (AhR), a ligand-activated transcription factor that integrates dietary and metabolic cues to control the complex transcriptional program[18,19]. Indeed, AhR overexpression leads to insulin resistance[20]. Conversely, germline *AhR* null mice have enhanced insulin sensitivity and improved glucose tolerance[21–26]. Moreover, mice that express a low-affinity AhR allele are less susceptible to obesity after exposure to a HFD and exhibit differences in fat mass, liver physiology, and hepatocyte gene expression compared to mice with high-affinity AhR[27]. However, the molecular mediators and mechanisms governing the diet-mediated association between AhR, PC lipid, and insulin pathway signaling in hepatocytes are largely unknown.

Here, our studies revealed that a HFD dramatically changes the lipid profile of intestinal epithelial exosomes from predominantly phosphatidylethanolamine (PE) to phosphatidylcholine (PC), which results in inhibition of the insulin response via binding of exosomal PC to AhR expressed in hepatocytes and suppression of genes essential for activation of the insulin pathway. Our results reveal a mechanism by which diet shapes the exosome lipid profile of intestinal epithelial cells to regulate liver/gut axis communication.

## Results

**HFD alters the composition of intestinal epithelial CD63$^+$A33$^+$ exosomes.** To study the effect of HFD on intestinal epithelial-released exosomes, we used a 12-month HFD-induced obesity mouse model (Supplementary Fig. 1a), which closely mimics human obesity with insulin resistance. HFD mice developed glucose intolerance (Supplementary Fig. 1b) and insulin resistance (Supplementary Fig. 1c) compared to mice fed a regular chow diet (RCD) for 12 months. The HFD mice were obese and had an increased body weight and epididymal white adipose tissue (WAT) weight (Supplementary Fig. 1d, e, respectively), plasma and liver triglycerides (Supplementary Fig. 1f, g, respectively), as well as fatty liver and liver steatosis (Supplementary Fig. 1h, i). Increased inflammatory cytokines were also noticed in the liver (Supplementary Fig. 1j) and intestinal tissue (Supplementary Fig. 1k) extracts of HFD-fed mice. Changes in the lipid profile of small intestinal tissues with increasing PC and acylcarnitine but not ceramide in HFD-fed mice (Supplementary Fig. 1l) were noticed.

Exosomes were isolated by differential centrifugation from the feces of a group of 12-month HFD-fed mice (H-Exo) and age- and sex-matched RCD mice (L-Exo). Sucrose-purified exosomes (Supplementary Fig. 2a) from RCD and HFD mice were characterized by transmission electron microscopy (Supplementary Fig. 2b). Exosome size was estimated by nanoparticle tracking analysis and the size ranges of L-Exo and H-Exo were $115 \pm 52$ nm and $120 \pm 54$ nm (Supplementary Fig. 2c), respectively. Both L-Exo and H-Exo exosomes were positive for CD63, CD81, CD9, MHCII, and A33 (intestinal epithelial cell marker) as assessed by western blot (Supplementary Fig. 2d). CD63 and A33 dual positivity was also assessed by confocal microscopy (Supplementary Fig. 2e). Mass spectrometry (MS)/MS analysis indicated that after fecal exosomes pulldown with CD63$^+$A33$^+$ antibody, no bacterial-derived proteins or bacterial EV proteins were detected (Supplementary Table 1). No 16S or 18S RNA bands were observed on an agarose gel in CD63$^+$A33$^+$ and CD63$^-$ EVs (Supplementary Fig. 2f). Quantitative polymerase chain reaction (qPCR) analysis further suggested that no bacterial 16S was amplified in CD63$^+$A33$^+$ exosomes, whereas no miR-375 (specifically encapsulated in host exosomes not in microbial EVs) was amplified in the CD63$^-$ EVs (Supplementary Fig. 2g, h, respectively). No detectable LPS (lipopolysaccharide), which is also a component of bacterial EVs[28] in either CD63$^+$A33$^+$ L-Exo or H-Exo exosomes, was detected (Supplementary Fig. 2i). Collectively, these results including the protein MS/MS profile, 16S and 18S RNA profile, and LPS lipid confirmed that exosomes used in the study were free from bacterial-derived products. Significantly higher absolute numbers of nanoparticles were isolated from HFD mice ($\sim 5 \times 10^{10}$ nanoparticles/g feces) than from RCD mice ($\sim 2 \times 10^{10}$ nanoparticles/g feces) (Supplementary Fig. 2j). Significantly higher numbers of CD63$^+$A33$^+$/g of feces were also found in HFD mice compared to RCD mice (Supplementary Fig. 2k), suggesting that the number of exosomes released from intestinal epithelial cells was affected by the HFD.

We next assessed whether HFD affected the composition of CD63$^+$A33$^+$ exosomes. While changes in protein (Supplementary Table 1) and miRNA expression were noted, (Supplementary Fig. 2l and Supplementary Table 2) the exosomal lipid profile was the most dramatically affected by the HFD (Fig. 1 and Supplementary Table 3). Triple-quadrupole MS analysis of total lipids revealed that L-Exo (collected at 6 months of feeding on RCD) was enriched in PE (56.8%), whereas H-Exo (collected at 6 months of HFD feeding) was enriched in PC, 22.9% (Fig. 1a). The concentrations of PE and PC in L-Exo were $86.3 \pm 1.76$ and $2.0 \pm 0.05$ nmol, respectively, whereas, in H-Exo, PE and PC concentrations were $13.4 \pm 1.68$ and $34.1 \pm 1.19$ nmol, respectively, using triple-quadrupole MS (Fig. 1b). Furthermore, 12 months after HFD feeding, the percentages of PC lipids in H-Exo dramatically increased to 38.8% and PE decreased from 10.6 to 1.3% (Fig. 1c). The concentrations of PE and PC in L-Exo were

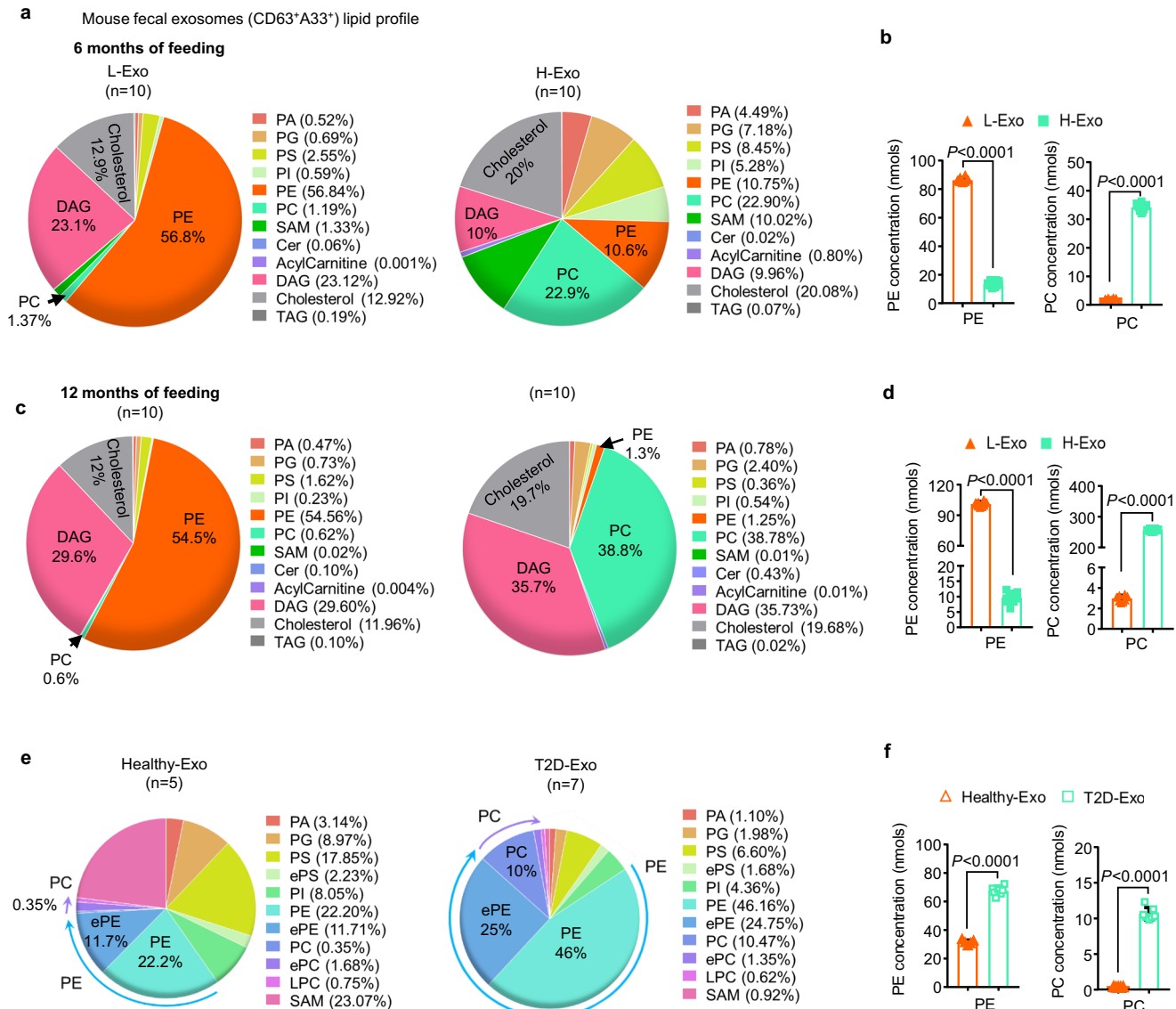

**Fig. 1 High-fat diet (HFD) alters the lipid composition of intestinal epithelial cell-released exosomes. a, b** Lipids were extracted from fecal exosomes (CD63⁺A33⁺) released from intestinal epithelial cells of mice fed either a regular chow diet (RCD) or a high-fat diet (HFD) for 6 months ($n = 10$/group). Pie charts representing the percentage of each lipid, as determined using triple-quadrupole mass spectrometry (**a**). PE and PC concentration (nmol) shown at the right panel (**b**). Filled triangle—L-Exo and filled rectangle—H-Exo. **c, d** Triple-quadrupole mass spectrometry of exosomal lipids harvested from mice feces after 12 months of their respective diets ($n = 10$/group). Pie charts representing the percentage of each lipid (**c**). PE and PC concentration (nmols) shown at the right panel (**d**). **e, f** Pie charts (**e**) representing the percentage of lipid subtypes derived from the fecal exosomes (CD63⁺A33⁺) of healthy (Healthy-Exo, $n = 5$) or patients with type 2 diabetes (T2D-Exo, $n = 7$). PE and PC concentration (nmols) determined by triple-quadrupole MS shown at the right panel (**f**). Hollow triangle—Healthy-Exo and hollow rectangle—T2D-Exo. Data are represented as mean ± S.D. Student's *t* test (two-tailed). Source data are provided as a Source Data file.

$101.0 \pm 2.40$ and $3.01 \pm 0.06$ nmols, respectively; however, in H-Exo, the PE and PC concentrations were $9.6 \pm 1.2$ and $256 \pm 2.9$ nmol, respectively, as determined by triple-quadrupole MS (Fig. 1d). Multiple correlation factor analysis revealed that PC was the most affected lipid. This result was also supported by the fact that the percentages of PC lipids in intestinal tissue of mice fed an HFD for 12 months were also increased (3.6%) compared with mice fed an RCD for 12 months (1.9%, Supplementary Fig. 1l). Notably, the L-Exo lipid composition was not significantly altered after 12 months vs. 6 months of RCD feeding. Triple-quadrupole MS analysis of fluorescence-activated cell sorting (FACS)-sorted A33⁺ cells from intestinal tissue indicates that HFD-fed mice also have higher PC than lean mice (Supplementary Fig. 3a and Supplementary Data 1). Collectively, these data suggest that the

higher PC is in the intestinal epithelial cells, then higher the PC will be in fecal CD63⁺A33⁺ exosomes. Next, we determined whether circulating plasma exosomes of mice fed the HFD also have higher concentrations of PC. In agreement with data generated from fecal CD63⁺A33⁺ exosomes, HFD-fed mice also have higher PC on circulating plasma exosomes (Supplementary Fig. 3b and Supplementary Data 1). Due to the sharper increase in fecal exosomal PC (RCD vs HFD feeding) from 0.6 to 38.8% when compared to plasma exosome PC that increases from 16.15 to 43.0%, we believe that fewer tissues are involved in releasing exosomes into the intestinal lumen when compared to blood exosomes. Based on the sharp increase in fecal exosomal PC and our primary goal of this study is to dissect out the role of PC in the insulin response, we used fecal exosomes throughout this study.

We next determined the dynamic changes in exosomal PE and PC levels using high-performance liquid chromatography (HPLC) analysis of $CD63^+A33^+$ exosomes from RCD and HFD mice at 3, 6, and 12 months of feeding with their respective diet. In agreement with the triple-quadrupole MS analysis, HPLC analysis also showed that H-Exo contained increased levels of PC (~40 μM and 80 μM at 6 and 12 months, respectively) compared to L-Exo (Supplementary Fig. 3c).

We next determined whether phosphatidylethanolamine N-methyl transferase (PEMT), the transferase enzyme that converts PE to PC, is increased in the intestinal tissue and liver of mice fed HFD vs. RCD controls. Western blot analysis indicated that over the time course of 3, 6, and 12 months of feeding on their respective diets, the levels of PEMT increased in intestinal tissue and liver of HFD mice compared to RCD mice (Supplementary Fig. 3d, e, respectively). Moreover, mouse colon epithelial (MC-38) cells treated with the fecal metabolites from HFD mice showed increased expression of the luciferase gene, driven by the PEMT promoter (Supplementary Fig. 3f), suggesting that the expression of the gene encoding PEMT is modulated by a HFD.

**The composition of intestinal epithelial $CD63^+A33^+$ exosomes is altered in insulin-resistant type II diabetes patients.** To determine whether these findings generated from obese mice can be applied to insulin-resistant type II diabetes, exosomes were isolated from stool samples of insulin-resistant patients with type II diabetes (T2D) and healthy subjects and were characterized. The size ranges of the exosomes derived from healthy subjects (Healthy-Exo) and exosomes derived from patients with diabetes (T2D-Exo) were $104 \pm 81$ nm and $190 \pm 86$ nm (Supplementary Fig. 4a), respectively. Like mouse exosomes, these human-derived exosomes were positive for CD63, CD9, CD81 (exosomal markers), and A33 (Supplementary Fig. 4b, c) and no 16S bacterial rDNA was detected (Supplementary Fig. 4d). Significantly higher numbers of total exosomes were found in the stool samples of patients with diabetes (~$4.5 \times 10^{13}$ nanoparticles/g feces) than in those from healthy subjects (~$2 \times 10^{13}$, Supplementary Fig. 4e). Absolute numbers of $CD63^+A33^+$ double-positive exosomes from T2D patients were also found significantly higher compared to healthy individuals (Supplementary Fig. 4f). Next, PE and PC from stool exosome samples of healthy and T2D patients were quantitatively analyzed using triple-quadrupole MS. Like the results from mouse exosomes, T2D patient exosomes also carried elevated levels of PC (~10%) compared to healthy individuals (~0.35%, Fig. 1e, Supplementary Table 3). The concentrations of PE and PC in Healthy-Exo were $31.3 \pm 0.95$ and $0.33 \pm 0.03$ nmol, respectively, whereas in T2D-Exo, PE and PC concentrations were $67.5 \pm 1.96$ and $10.6 \pm 0.60$ nmol, respectively, determined by triple-quadrupole MS (Fig. 1f). In agreement with the triple-quadrupole MS analysis, HPLC quantitative analysis also showed elevated levels of PC (~22 μM) in T2D Exo (Supplementary Fig. 4g, h), whereas in healthy Exo, PC was undetectable. Abnormally high and low cellular PC/PE ratios are linked to T2D disease progression[29]. Although the absolute value of PE from stool exosome samples of T2D patients was increased (Supplementary Fig. 4h), ratios of PC/PE were found much higher in T2D exosomes than in healthy exosomes (Supplementary Fig. 4i). Similar results were obtained from mice exosomes and small intestinal tissues (Supplementary Fig. 4j, k).

**H-Exo and T2D-Exo contribute to the development of insulin resistance and glucose intolerance.** We next investigated whether the altered lipid profile of H-Exo plays a role in the response to glucose and insulin sensitivity. To assess the in vivo effect of H-Exo, lean mice were given either $CD63^+A33^+$ L-Exo or H-Exo

($2 \times 10^9$/dose in 200 μl of PBS with 3, 5, 10, or 14 doses) by gavage every day for 14 days while being fed an HFD. H-Exo (at any dose) had no effect on the bodyweight of mice over the 14-day treatment period (Suppl Fig. 5a), but 10 doses of H-Exo resulted in a dose-dependent impairment in insulin sensitivity (Supplementary Fig. 5b, 10-dose panel). Fourteen doses of H-Exo resulted in further impairments in insulin sensitivity, as mice showed insulin resistance at all time points after insulin injection (Supplementary Fig. 5b; 14-dose panel). Based on this result, 14 doses (1 dose/day) of L-Exo or H-Exo were given for all mouse experiments throughout the study. Glucose tolerance (GTT) and insulin tolerance testing (ITT) suggested that H-Exo recipient mice become glucose intolerant and showed the impression of insulin resistance (Fig. 2a).

The comprehensive lab animal monitoring system (CLAMS, metabolic cages) assay suggested that H-Exo administration had no effect on recipient mice body composition (weight, fat, and lean mass), energy expenditure, oxygen consumption, $CO_2$ production, respiratory exchange ratios, locomotor activity, and ambulatory activity compared to PBS and L-Exo recipient mice (Supplementary Fig. 5c–i). H-Exo recipient mice exhibited less food intake compared to PBS and L-Exo recipient mice (Supplementary Fig. 5j).

The hyperinsulinemic–euglycemic clamp assay revealed that mice receiving H-Exo for 14 days showed a significantly lower glucose infusion rate (GIR) as compared to mice receiving L-Exo during 80–120 min of the clamp assay (Fig. 2b). Moreover, H-Exo recipient mice showed higher blood glucose levels compared to L-Exo recipient mice (Fig. 2c). However, a significant difference was observed in insulin levels among the groups at basal time points but not during the clamp assay time points (Fig. 2d; Supplementary Data 2 for insulin concentration during GTT). Endogenous hepatic glucose production was observed significantly higher and percent suppression during the clamp assay was significantly lower in H-Exo recipient mice compared to PBS and L-Exo recipient mice (Fig. 2e, f). Furthermore, the rate of whole-body glycolysis was found significantly inhibited in H-Exo recipient mice (Fig. 2g). Individual tissue glucose uptake data suggested that the uptake of brown adipose tissue, WAT, and skeletal muscle (gastroc) tissue of H-Exo recipient mice had significantly less glucose uptake compared to PBS or L-Exo recipient mice (Fig. 2h). Plasma free fatty acids (FFA) at basal level (before clamp, −10 min) were not found affected, but at 120 min, they were significantly elevated in plasma of the H-Exo recipient mice (Fig. 2i). Overall, these results showed that H-Exo causes glucose intolerance and insulin resistance in mice.

Next, we determined whether $CD63^+A33^+$ T2D-Exo from the stool of T2D human patients might elicit glucose intolerance and insulin resistance in C57BL/6 SPF mice. Mice were gavage-given human $CD63^+A33^+$ (Healthy-Exo or T2D-Exo) exosomes for 14 days while being fed HFD. T2D-Exo had no effect on the bodyweight of mice over the 14-day treatment period (Supplementary Fig. 5k). While the effects were not as pronounced as those observed after H-Exo treatment, GTT and ITT results suggested that mice that received T2D-Exo developed glucose intolerance and insulin resistance (Fig. 2j).

To determine whether the deleterious effect of H-Exo on the insulin response is dependent on the gut microbiome, age-matched germ-free C57BL/6 male mice were orally administered H-Exo or L-Exo and Healthy-Exo or T2D-Exo for 14 days while being fed the HFD. Surprisingly, H-Exo treatment led to both SPF and germ-free mice to develop glucose intolerance and reduced their response to insulin (Fig. 2k). These results suggest that H-Exo and T2D-Exo have a negative impact on insulin response that is independent of the gut microbiome. To gain insight into whether the pathophysiological effect of H-Exo can be altered in a gut

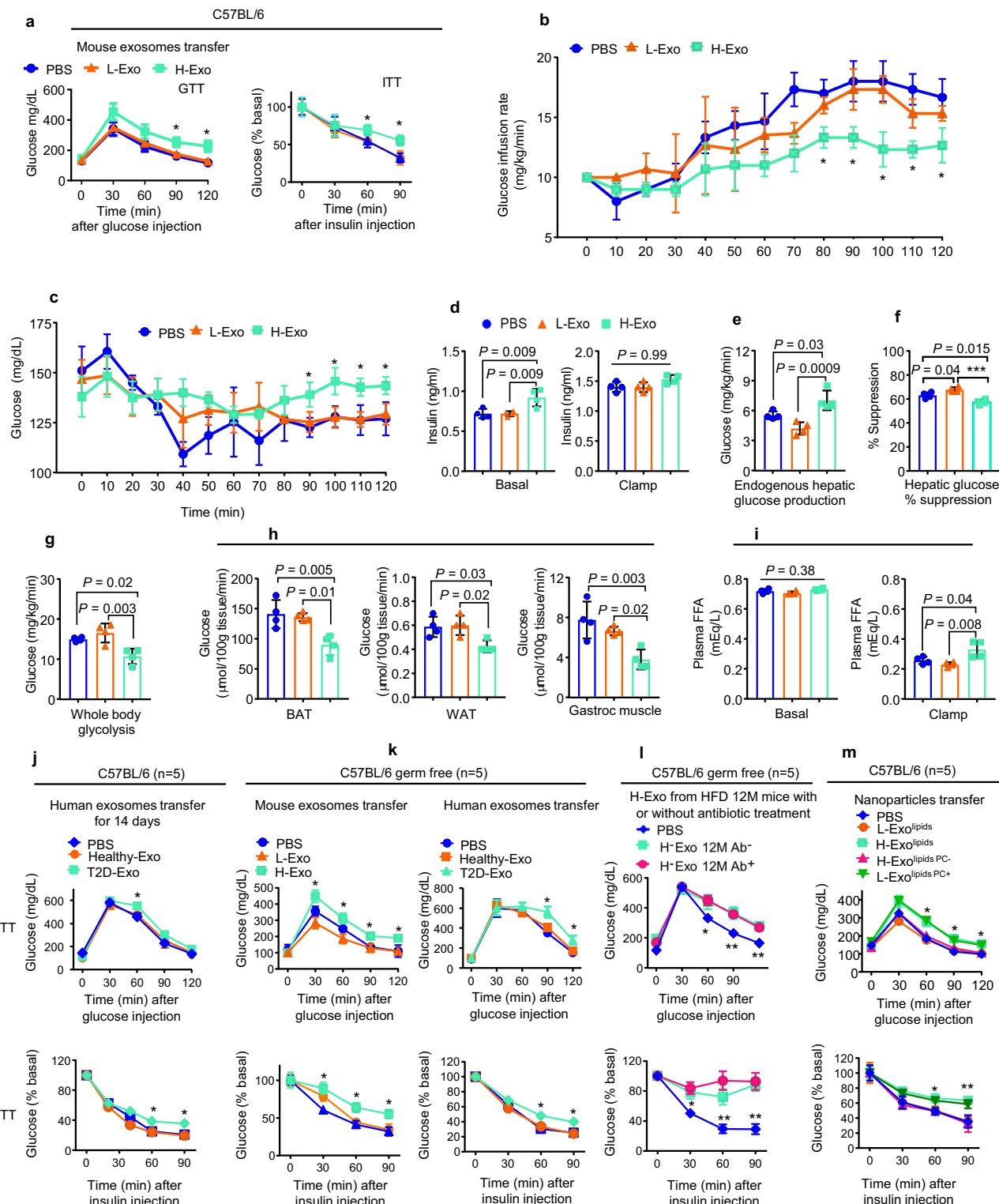

microbiota-dependent manner, 12-month HFD-fed mice were treated with/without antibiotics for 7 days and then H-Exo was isolated from antibiotic-treated mice (H-Exo 12 M Ab$^+$ and H-Exo 12 M Ab$^-$) and gavage-given to germ-free mice to test the insulin response. No significant difference in glucose intolerance and insulin resistance was observed in both groups of germ-free H-Exo recipient mice (Fig. 2l), suggesting that changes in bacterial composition due to antibiotic treatment do not seem to alter the function of H-Exo in terms of induction of glucose intolerance and insulin resistance as compared to PBS-treated mice.

Since we observed dynamic changes in PC levels in H-Exo, we explored the role of exosomal lipids in regulating insulin sensitivity. To this end, nanoparticles were generated from total lipids of both L-Exo and H-Exo ($2 \times 10^9$ exosomes). To generate H-Exo-derived nanoparticles with depleted PC, total lipids from exosomes were extracted with chloroform and separated via thin-layer chromatography (TLC) (Supplementary Fig. 5l) and the PC band was then depleted from H-Exo (H-Exo$^{\text{lipids PC−}}$), while an equal amount of commercial PC (40 nmols) was added to L-Exo (L-Exo$^{\text{lipids PC+}}$). The band containing PC (Supplementary Fig. 5l,

**Fig. 2 CD63⁺A33⁺ exosomes derived from HFD mice induce insulin resistance in mice fed an HFD. a** Glucose (GTT) and insulin tolerance tests (ITT) for C57BL/6 mice ($n = 10$) after receiving adoptive transfer of mouse CD63⁺A33⁺ fecal exosomes for 14 days while mice were fed HFD. Filled circle—PBS, filled triangle—L-Exo, and filled rectangle—H-Exo. **b** Glucose infusion rate (GIR) during the clamp assay ($n = 4$/group). **c** Blood glucose levels during the clamp assay ($n = 4$/group). **d** Plasma insulin levels at basal (−10 min) and during the clamp assay (120 min). **e, f** Hepatic glucose production and % suppression of hepatic glucose production ($n = 4$/group). **g** Whole-body glycolysis rate determined by the clamp assay ($n = 4$/group). **h** Glucose uptake by brown adipose (BAT), white adipose tissue (WAT), and muscle tissue after the clamp assay ($n = 4$/group). **i** Plasma free-fatty acids at basal and during the clamp assay ($n = 4$/group). **j–m** GTT (upper panels) and ITT (lower panels) for either C57BL/6 or C57BL/6 germ-free mice after receiving adoptive transfer of human exosomes (**j**), mouse and human exosomes (**k**), mouse exosomes with or without antibiotic treatment (**l**), and nanoparticles from L-Exo or H-Exo with added (PC+) or depleted PC (PC−) for 14 days while mice were fed HFD ($n = 5$/group). Data are represented as the mean ± SD. One-way ANOVA with a Tukey post hoc test. * < 0.05; ** < 0.01, and *** < 0.001. Statistical significances were shown between the PBS and H-Exo group or as otherwise indicated. Source data are provided as a Source Data file.

highlighted in the red rectangle) was identified based on standard PC migration in TLC and then removed. The PC band and remaining lipids from H-Exo lipid TLC were profiled by triple-quadrupole MS (Supplementary Data 3). These lipid nanoparticles were orally administered to HFD-fed mice for 14 days. Again, the nanoparticles had no effect on the body weight of mice over the 14-day treatment period (Supplementary Fig. 5m), but GTT and ITT results showed that mice receiving H-Exo lipids or L-Exo lipids^PC+ developed glucose intolerance and insulin resistance compared to mice receiving L-Exo lipids or H-Exo lipids^PC− (Fig. 2m). Although diacylglycerols, ceramides, and acylcarnitines have been reported to contribute to insulin resistance[30], the band containing PC had no detectable level of those lipids. The level of ceramides, diacylglycerols, and acylcarnitines in H-Exo and L-Exo has no significant changes as pronounced PC levels and remaining lipids were not affected due to PC removal (Supplementary Fig. 5n–p, respectively). These results suggest that HFD-induced elevations in exosomal PC contribute to insulin resistance and glucose intolerance. Since nanoparticles made from total lipids extracted from H-Exo have a similar effect as H-Exo on insulin-mediated glucose uptake by hepatocytes (Supplementary Fig. 5q), the effects of H-Exo proteins and RNAs in insulin resistance were not further investigated in this study.

**CD63⁺A33⁺ exosomal PC modulates the targeting of exosomes to liver cells.** In order to examine how intestinal epithelial cell-derived exosomes modulate gut–liver communication, we began by establishing a mouse colon epithelial cell line (GFP-MC38), which releases GFP-positive exosomes (Supplementary Fig. 6a). Exosome size was found to be $110 \pm 45$ nm (Supplementary Fig. 6b, c). These exosomes were positive for CD63, CD9, and A33 (Supplementary Fig. 6d). We next injected GFP-MC38 cells ($5 \times 10^5$) into the colon of C57BL/6 mice. Six weeks post injection, tumor was visible, and mice were sacrificed, and organs were harvested. Visualization of the liver with confocal microscopy revealed that the injected GFP-MC38 exosomes indeed reached the liver (Fig. 3a).

Next, we determined whether endogenous gut epithelial exosomes administered via oral gavage have similar trafficking routes as colon-injected GFP-MC38 exosomes, by double-labeling H-Exo and L-Exo ($2 \times 10^9$ exosomes) with DIR and PKH-26 fluorescent dyes for live mouse imaging at multiple time points (Supplementary Fig. 7a). Within 30 min after oral administration, DIR-labeled exosomes were detected in the peripheral blood, peaked at 6 h, and then started decreasing by 48 h (Supplementary Fig. 7b); DIR exosomes went undetected in the circulation by 96 h post oral administration (Supplementary Fig. 7c). Forty-eight hours after oral administration of H-Exo or L-Exo, mice were sacrificed, and organs were collected for imaging (Supplementary Fig. 7d). Scanned organ images suggested that labeled H-Exo or L-Exo trafficked to the liver with the strongest signal when compared with the signals detected in other organs. The confocal

imaging analysis indicated that intestinal epithelial cell exosomes trafficked to the liver via the portal vein but not through the lymphatic system (Supplementary Fig. 8a, b, respectively). DIR-labeled PC nanoparticles were gavage given to mice, and after 6 h, DIR signals were measured. The results suggested that ~60% of the nanoparticles reached the liver during the 6-h period after gavaging (Supplementary Fig. 8c). At the molecular level, triple-quadrupole MS analysis revealed that liver PC, TAGs, DAGs, and fatty acyl chains in TAGs increased, and ceramides were found to decrease (Supplementary Fig. 8d).

We then sought to identify the exosome recipient cells. Liver cells of the mice imaged as described above were then isolated and stained with anti-albumin (a marker for hepatocytes) and F4/80 (a marker for macrophages or Kupffer cells) and analyzed by flow cytometry. Flow cytometry analysis revealed that the majority of L-Exo (>80%) was taken up by hepatocytes (Fig. 3b, L-Exo panel) and far fewer (~12%) by F4/80-positive macrophages. By contrast, the majority of H-Exo (>64%) was taken up by F4/80-positive macrophages compared to ~39.5% taken up by hepatocytes (Fig. 3b, H-Exo panel). These results agree with those generated with confocal microscopy (Fig. 3c). It was observed that ~88% H-Exo gain entry via phagocytosis was found to be inhibited by cytochalasin D (Fig. 3d) compared to other inhibitors. The uptake of L-Exo and H-Exo was further confirmed by confocal microscopy in primary hepatocytes and Kupffer cells (Supplementary Fig. 9a, b). Kupffer cells cultured with H-Exo in the presence of cytochalasin D showed inhibition of uptake of H-Exo (Supplementary Fig. 9c).

Further, confocal imaging of cell lines showed that hepatocytes (human HepG2 and mouse FL83B cells) preferentially take up L-Exo compared to H-Exo, whereas human monocytes (U937 cells) preferentially take up H-Exo compared to L-Exo (Supplementary Fig. 9d–f). Since H-Exo is enriched with PC, we investigated whether exosomal PC plays a role in mediating the preferential uptake of exosomes by specific cell types by studying the effects of PC depletion from H-Exo. Nanoparticles were prepared with either a depleted band containing PC (PC–) or were supplemented with a known amount of (40 nmol) synthesized PC (PC+). Nanoparticles were then labeled with PKH26 dye and cocultured with primary hepatocytes and Kupffer cells for 16 h. Flow cytometry results (Fig. 3e) generated following administration of L-Exo^lipids PC+ indicated that the addition of PC lipids to L-Exo, leads to a significant reduction in exosome uptake by primary hepatocytes (from 88.8 to 27.8%), whereas the removal of PC from H-Exo lipids (H-Exo^lipids PC−) increased their uptake by primary hepatocytes (from 40.8 to 72.9%). In Kupffer cells (Fig. 3f), removal of PC from H-Exo led to a reduction in exosome uptake (from 72.3% to 19.2%), and the addition of PC lipid to L-Exo led to an increase in their uptake (from 16.8 to 67.8%). Therefore, these results suggest that the uptake of CD63⁺A33⁺ exosomes by specific liver cell types is dependent on their lipid composition, specifically the percentage of PC lipids they contained.

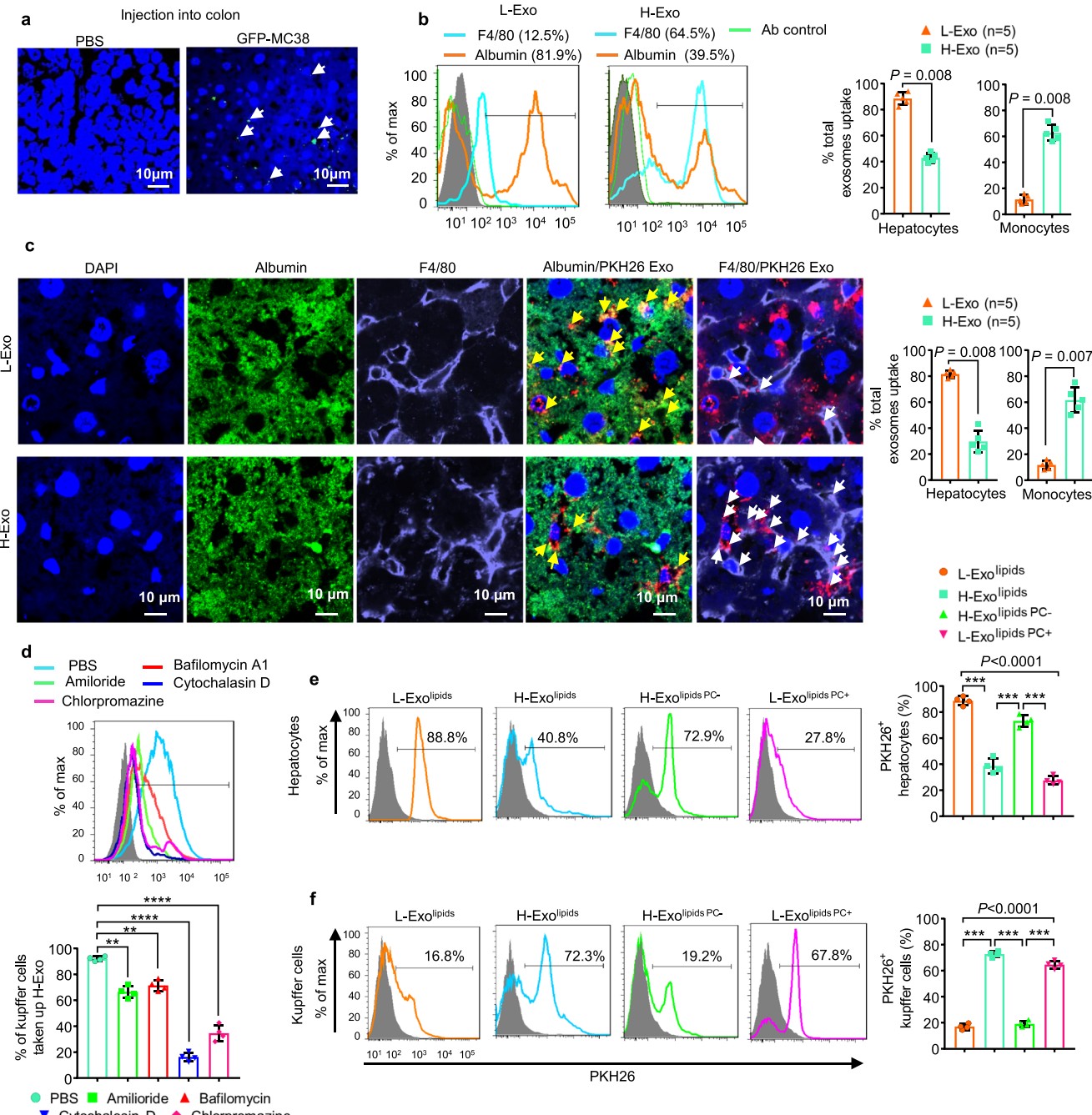

**Fig. 3 Uptake of CD63+A33+ exosomes by liver cells depends on exosomal lipid composition. a** Confocal images of GFP-positive exosomes detected in mouse liver after injection of GFP-MC38 epithelial cells into the colon. DAPI was used to stain the nucleus. Each data point was measured in triplicate. **b** Flow cytometry analysis of PKH-26-labeled exosome uptake by hepatocytes (albumin+) and Kupffer cells (F4/80+). Filled triangle—L-Exo and filled rectangle—H-Exo. **c** PKH26-labeled exosomes visualized by confocal microscopy in hepatocytes/albumin+/green (yellow arrow) and Kupffer cells/F4/80/purple (white arrow) (left). The percentages of total exosome uptake per cell type are summarized on the right. Scale bar as indicated. Each data point was measured in triplicate. **d** Flow cytometry analysis of Kupffer cells cultured with PKH26-labeled H-Exo for 16 h in the presence or absence of endocytosis inhibitors (indicated). Percentage of PKH26+ Kupffer cells summarized below. **e**, **f** PKH-26-labeled nanoparticles were cultured with primary hepatocytes (upper panels, **e**) and Kupffer cells (lower panels, **f**). Cells were analyzed by flow cytometry and the percentage of PKH-26-positive cells were assessed after treatment with each nanoparticle summarized at the right. Circle—nanoparticles made up of total lipids of L-Exo; rectangle—H-Exo lipids; upward triangle—H-Exo lipid-depleted PC and downward triangle—L-Exo PC supplemented. Source data are provided as a Source Data file. Data are represented as the mean ± SD. Student's t test (two-tailed) or one-way ANOVA with a Tukey post hoc test. ** < 0.01; *** < 0.001, and **** < 0.0001.

Once targeting of the liver by CD63+A33+ exosomes was confirmed, we evaluated whether this had an effect on glucose homeostasis. In particular, the effect of exosomes on glucose uptake by primary hepatocytes was evaluated. Inhibition of glucose uptake was observed in primary hepatocytes treated with

H-Exo compared to L-Exo-treated hepatocytes (Suppl Fig. 9g). Since the hyperinsulinemic–euglycemic clamp assay suggested that H-Exo administration affected liver, WAT, and muscle tissues, these results were further confirmed in vitro in mouse hepatocyte (FL83B cells), adipocyte (3T3−L1), and myocyte

(C2C12) cell lines. A 16-h treatment of H-Exo or T2D-Exo of these cell lines exhibited inhibition of glucose uptake (Supplementary Fig. 9h–j). Furthermore, FL83B cells treated with H-Exo, H-Exo[lipids], and L-Exo[lipids PC+] took up less glucose compared with L-Exo and H-Exo[lipids PC−]. These results suggest that H-Exo and H-Exo[lipids] inhibit the glucose uptake in mouse primary hepatocytes (Supplementary Fig. 9k).

**H-Exo-activated macrophages increase the production of TNF-α and IL-6 in the plasma, liver, and adipose tissue.** Pro-inflammatory cytokines contribute to insulin resistance in the liver by inhibiting insulin signal transduction[31–33]. Next, we investigated whether exosomes have an effect on the cytokine profile. H-Exo vs L-Exo treatment resulted in the induction of an array of inflammatory cytokines detected in the plasma and WAT, including TNF-α and IL-6, which are known to contribute to insulin resistance (Fig. 4a, b; Supplementary Data 4). Increased TNF-α and IL-6 levels following H-Exo treatment were further confirmed by ELISA in plasma (Fig. 4c). In the context of the role of fecal CD63+A33+ exosomes isolated from mice fed HFD vs. RCD in terms of cytokine production, germ-free mice-fed CD63+A33+ exosomes for 14 days had TNF-α and IL-6 in intestinal and liver tissue quantified. The results suggested that H-Exo induced TNF-α and IL-6 in the liver (Supplementary Fig. 10a); however, no significant differences were found in intestinal tissues (Supplementary Fig. 10b).

Taken together, our findings suggested that the uptake of H-Exo by liver macrophages may result in macrophage activation and subsequent release of TNF-α and IL-6, thus contributing to the development of insulin resistance. To determine whether macrophages play a role in H-Exo-mediated insulin resistance, macrophages were depleted in mice treated with H-Exo. The effectiveness of the depletion was confirmed by flow cytometry by F4/80 staining of whole blood (Supplementary Fig. 10c). Depletion of macrophages led to a reduction in TNF-α and IL-6 levels following treatment with H-Exo vs. L-Exo (Supplementary Fig. 10d). Moreover, insulin sensitivity was improved in H-Exo-treated mice with depleted macrophages compared with H-Exo-treated mice without depletion of macrophages (Fig. 4d), although the improvement was found to be significantly different at only one-time point (at 60 min after insulin injection). These results suggested that the macrophage cytokines released in response to H-Exo may partially contribute to insulin resistance.

Macrophage activation plays a pathogenic role in hepatic insulin resistance[34,35]. Next, we explored the effect of cytokines from H-Exo- or L-Exo-treated macrophages on hepatocyte glucose uptake. First, we determined the minimum concentration of H-Exo that caused the inhibition of hepatocyte glucose uptake. The glucose uptake assay results suggested that a minimum dose of $5 \times 10^5$ H-Exo was required to significantly inhibit glucose uptake (Fig. 4e). Adding the supernatant from macrophage cultures treated with an elevated dose of H-Exo to hepatocytes further decreased their glucose uptake. However, no reduction in glucose uptake was observed when the supernatant from macrophages treated with L-Exo was added to the hepatocytes (Supplementary Fig. 10e). Furthermore, the nanoparticles generated from H-Exo total lipids (H-Exo[lipids]) and synthesized PC (34:2) (Supplementary Fig. 10f) had similar impacts on glucose uptake as did H-Exo (Fig. 4f). Neutralizing both TNF-α and IL-6 (Fig. 4g) in the macrophage supernatants improved glucose uptake, suggesting that macrophage-derived IL-6 and TNF-α play an additive role with H-Exo in inhibiting glucose uptake in hepatocytes.

**AhR involved in H-Exo-induced insulin resistance.** AhR is a ligand-activated transcription factor that integrates dietary and metabolic cues to control transcriptional programs, including the insulin-signaling pathway in hepatocytes[15–24]. We began by investigating whether H-Exo alters the expression of AhR in mouse livers, and indeed our Affymetrix (GSE156848) and qPCR results indicated that the gene encoding *AhR* is upregulated following H-Exo gavaging (Fig. 5a). The induction of AhR protein was further confirmed by western blot analysis (Fig. 5b). Furthermore, H-Exo and nanoparticles made from H-Exo lipids (H-Exo[lipids]) induced AhR protein in mouse hepatocytes (Fig. 5c). Induction of AhR in FL83B cells was also confirmed in a PC-dose-dependent manner (Fig. 5d). The glucose uptake in mouse hepatocytes treated with PC (34:2) was inhibited in a PC-dose-dependent manner also (Fig. 5e). These results suggest that PC-mediated inhibition of hepatic glucose uptake is associated with induction of AhR.

We next studied whether PC binds to AhR. We used surface plasmon resonance (SPR) to test this hypothesis. H-Exo-derived total lipids (Fig. 5f) and PC (34:2) lipid (Fig. 5g) were immobilized on a LIP-1 sensor. Recombinant AhR protein was prepared and run over the immobilized lipids (nanoparticles). As shown in Fig. 5f, g, the sensorgram of SPR peaks revealed that the AhR protein interacts with H-Exo lipids and PC (34:2) lipid nanoparticles, but not L-Exo lipid nanoparticles (Supplementary Fig. 11). Furthermore, PC (34:2) lipid was coated onto an ELISA plate and incubated with recombinant AhR protein and subsequently detected by anti-AhR antibody (Fig. 5h). Then, PC binding to AhR was demonstrated by immobilizing recombinant AhR on the NTA chip (protein sensor). H-Exo[lipids PC−] and PC (34:2)[lipid] were run over the immobilized AhR (Fig. 5i). We further explored whether H-Exo treatment also resulted in increased AhR activation (phosphorylation). The confocal imaging data show that PKH26-labeled H-Exo was co-localized with exosomal marker CD63 (Supplementary Fig. 12a). The specificity of co-localization of exosomes with AhR was further demonstrated by the fact that PKH26-labeled H-Exo (Supplementary Fig. 12b) led to co-localization of H-Exo with AhR in the cytosol of hepatocytes, suggesting that exosomes interact with cytosol AhR. Simply mixing an equal amount of PKH26 dye with H-Exo (Supplementary Fig. 12c) did not lead to co-localization with AhR. AhR was increased in the nucleus of H-Exo-treated hepatocytes (Fig. 5j). Further, confocal images of hepatocytes showed that phosphorylated AhR (pAhR) increased in the nucleus after H-Exo treatment, while no induction of pAhR was observed in L-Exo and PBS treatment (Supplementary Fig. 13a). AhR activation was also indicated by the fact that induction of AhR downstream targets genes (*Cyp1a1*, *Cyp1a2*, and *Cyp1b1*) by H-Exo (Supplementary Fig. 13b). The western blot analysis data generated from hepatocyte cell lines were reproduced with primary hepatocytes (Supplementary Fig. 13c).

Liver macrophages also take up H-Exo, so we then determined whether the expression of AhR in liver macrophages is induced upon uptake of H-Exo. qPCR analysis of the expression of *AhR*, *TNF-α*, and *IL-6* in liver macrophages indicated that increases of *TNF-α* and *IL-6* preceded induction of AhR (3 h, 6 h vs. 12 h). This result suggested that the expression of AhR is induced but the induction of AhR is secondary to the activation of macrophages by H-Exo (Supplementary Fig 14a–c).

Using AhR-deficient ($AhR^{−/−}$) mice, we next examined the role of AhR in H-Exo-mediated insulin resistance. In $AhR^{−/−}$ mice, H-Exo did not impair glucose tolerance or insulin responsiveness, unlike its effects in wild-type C57BL/6 mice (Fig. 5k). No difference in body weight was seen between $AhR^{−/−}$ mice and wild-type C57BL/6 mice (Supplementary Fig. 14d), suggesting that prevention of H-Exo- mediated insulin resistance in $AhR^{−/−}$ mice are unlikely due to changes in body weight. Consistent with these results, there was no difference in glucose

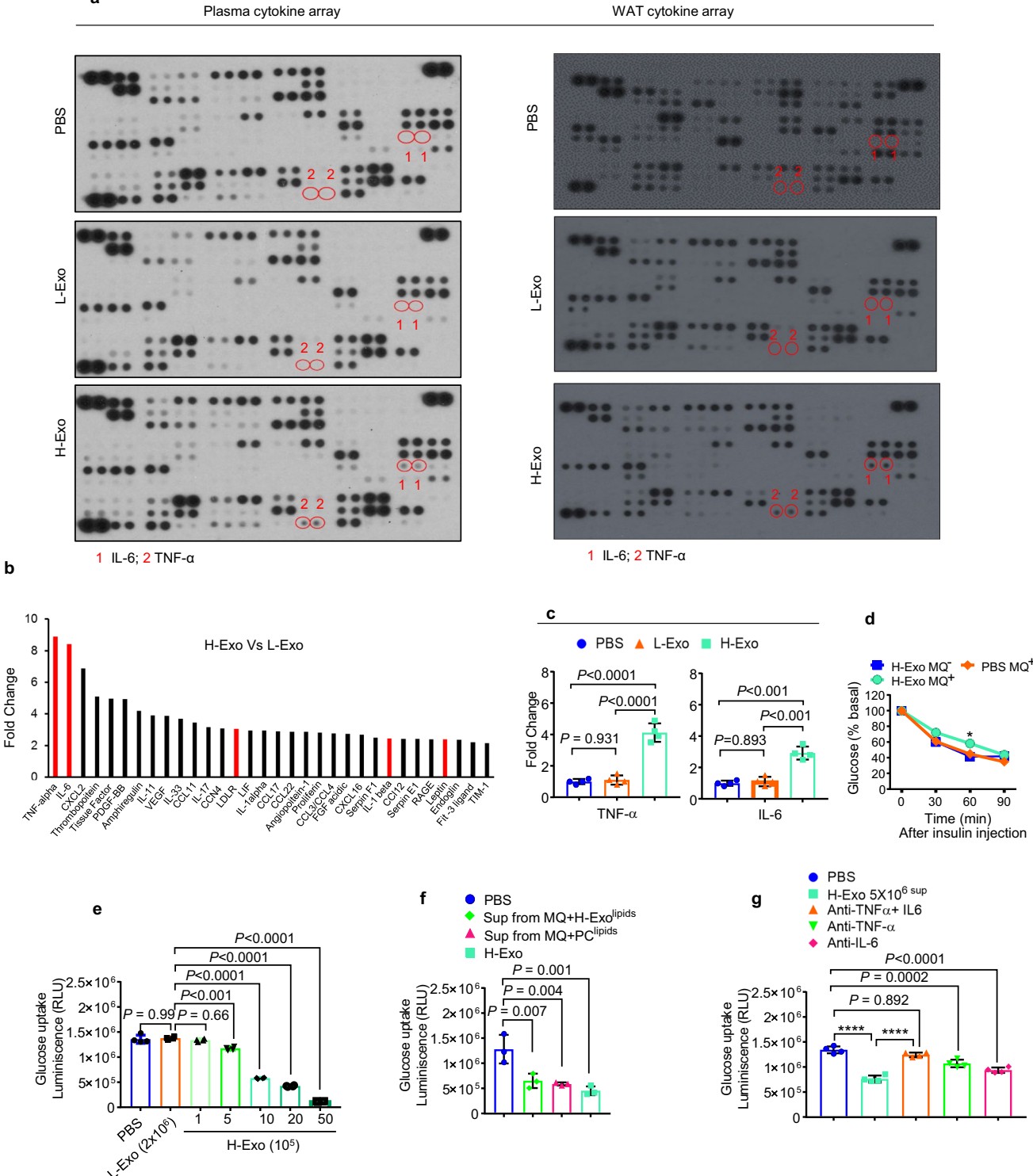

**Fig. 4 Crosstalk between hepatocytes and macrophages contributes to insulin resistance. a** Plasma and WAT cytokine array of mice that received CD63+A33+ exosomes (L-Exo or H-Exo) for 14 days. **b** Fold change in H-Exo vs. L-Exo-induced plasma cytokine expression for all cytokines showing greater than two-fold change. Red bars show cytokines/factors known to be involved in insulin resistance. **c** TNF-α (left) and IL-6 (right) upregulation following treatment with H-Exo were confirmed by ELISA in plasma. Filled circle—PBS, filled triangle—L-Exo, and filled rectangle—H-Exo. **d** ITT performed on C57BL/6 mice that received exosomes via adoptive transfer for 14 days followed with or without macrophage depletion. Filled rectangle—macrophage-depleted mice treated with H-Exo; filled diamond—mice without macrophage depletion treated with PBS and circle—mice without macrophage depletion treated with H-Exo. **e** Glucose uptake assay performed on hepatocytes cultured with different concentrations of H-Exo (as indicated in the figure). **f** Glucose uptake assay performed on mouse hepatocytes supplemented with supernatant derived from macrophages cultured with nanoparticles derived from H-Exo total lipids (H-Exo Nano) and PC (34:2). **g** Supernatants from H-Exo-treated macrophages (monocytes+ 5 × 10⁶) were preneutralized with anti-TNF-α and/or anti-IL-6 antibodies. Glucose uptake by hepatocytes cultured in the presence of preneutralized supernatant was estimated. Data are represented as the mean ± SD. One-way ANOVA with a Tukey post hoc test. * < 0.05; **** < 0.0001. Source data are provided as a Source Data file.

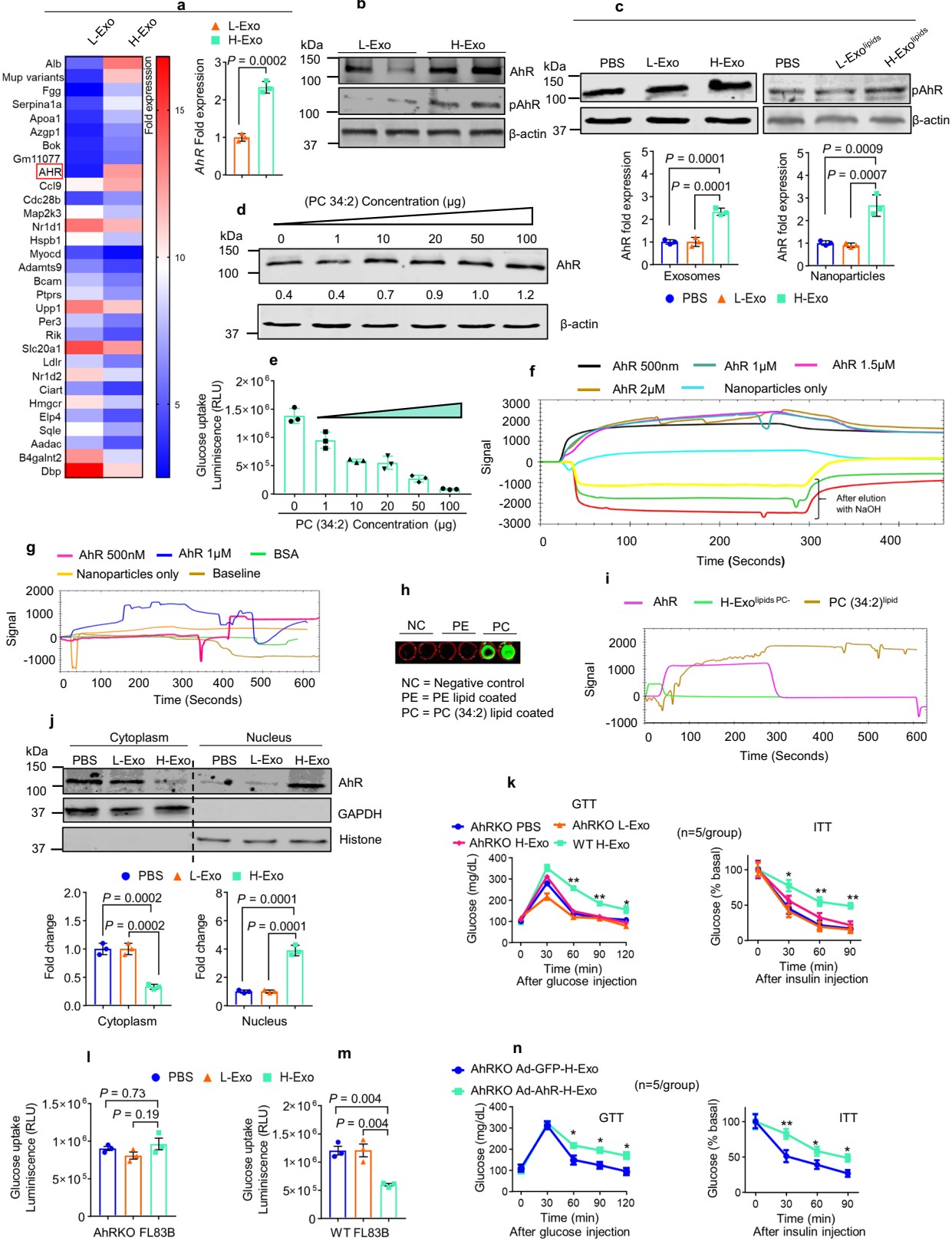

uptake in $AhR^{-/-}$ mouse hepatocytes treated with H-Exo vs L-Exo unlike C57BL/6 mice hepatocytes (Fig. 5l, m). To further demonstrate the role of hepatocyte AhR in H-Exo-mediated insulin resistance, $AhR^{-/-}$ mice were tail vein injected with Ad-AhR or Ad-GFP ($5 \times 10^9$/mouse in 200 µl of PBS). AhR protein was detected in liver tissue of mice treated with Ad-AhR but not

in Ad-GFP (Supplementary Fig. 14e) on day 7 after the injection. Starting day 3 after the injection, mice that were gavage-given H-Exo for 14 days developed glucose intolerance and had a reduced response to insulin (Fig. 5n). These results suggest that H-Exo PC contributes to insulin resistance via overexpression and subsequent activation of AhR.

**Fig. 5 HFD-induced CD63$^+$A33$^+$ exosomal lipids contribute to insulin resistance in an AhR-dependent manner. a** Representative gene expression heat map for the Affymetrix array of liver tissue from mice orally administered exosomes for 14 days. Induction of *AhR* expression highlighted in the red box. Elevated *AhR* expression was confirmed by qPCR (bar graph, right, *n* = 3). Filled triangle—L-Exo and filled rectangle—H-Exo. **b** Total AhR protein expression was confirmed by western blots in liver tissue. **c** Phosphorylated AhR (pAhR) protein expression in hepatocytes (FL83B cells) cultured with L-Exo, H-Exo, L-Exo$^{lipids}$, or H-Exo$^{lipids}$. Densitometry analysis summarized below. Filled circle—PBS, filled triangle—L-Exo, and filled rectangle—H-Exo. **d** FL83B cells were cultured with different concentrations (as indicated) of PC (34:2) for 16 h, and the resulting effects on AhR expression were determined by western blots. Ratio to β-actin shown in the middle as numbers. **e** Glucose uptake assay for FL83B cells cultured with varying concentrations of PC (34:2). **f, g** SPR sensogram showing the interaction of AhR recombinant protein with nanoparticles derived from total lipids of H-Exo (**f**) and PC (34:2) (**g**). **h** PC direct binding to AhR protein. **i** SPR was performed with AhR protein-coated onto an NTA chip and H-Exo$^{lipid\ PC-}$ and PC (34:2)$^{lipid}$ run over as the mobile phase. **j** AhR expression in the cytoplasm vs. nucleus of mouse hepatocytes cultured with L-Exo or H-Exo. Densitometry analysis of cytoplasmic (left) vs. nuclear (right) AhR protein expression following treatment with L-Exo or H-Exo summarized below. **k** GTT and ITT performed on AhR null (AhR$^{-/-}$) HFD-fed mice treated with CD63$^+$A33$^+$ exosomes (L-Exo or H-Exo) for 14 days. **l, m** Glucose uptake assay performed on AhR-knockout (AhRKO, **l**) and wild-type (**m**) FL83B cells. **n** GTT and ITT of *AhR$^{-/-}$* mice with re-expression of AhR in hepatocytes by adenovirus (5 × 10$^9$ pfu) injected via tail vein. Mice were orally gavaged with H-Exo for 14 days while being fed the HFD. Student's *t* test, one-tailed. Data represent the mean ± SD. Student *t* test (two-tailed) or one-way ANOVA with a Tukey post hoc test. * < 0.05 and ** < 0.01. Source data are provided as a Source Data file. Each data point was measured in triplicate.

**H-Exo administration affects insulin signaling and causes insulin resistance.** Insulin resistance is characterized by impaired insulin signaling in multiple metabolic organs, including liver, adipose tissue, and skeletal muscle[36]. Since the clamp assay (see Fig. 2b–i) revealed that insulin responses were impaired in all three major metabolic organs (liver, WAT, and muscle) in H-Exo recipient mice, we further quantified the expression of genes that regulate the insulin pathway in those metabolic organs using qPCR array. H-Exo treatment led to increasing the expression of *Ptpn1*, *Igfbp1*, *Jun*, and *Ldlr* and decreasing the expression of *IRS-2*, *IRS-1*, *Pparg*, and *Srebf1* (Fig. 6a, b) in the liver compared with L-Exo-treated mice. Cytokine array analysis also indicates that H-Exo treatment led to increasing the expression of TNF-α and IL-6 and decreasing the expression of IL-10 (Supplementary Fig. 15a) in the WAT tissue compared with L-Exo-treated mice. Among the expression of genes affected by H-Exo, downregulation of *IRS-1* and *IRS-2* as a result of H-Exo treatment has been demonstrated in WAT and muscle tissue (Supplementary Fig. 15b, c). Inhibition of activation of IRS-2 was indicated by measuring pIRS-2 levels by western blot analysis in tissue extracts, including liver, WAT, and muscle (Fig. 6c). Unlike wild-type B6 mice, *AhR$^{-/-}$* mice gavage-given H-Exo did not result in the inhibition of expression of IRS-2 in the liver, adipose tissue, and muscle tissue (Fig. 6d), indicating that H-Exo-mediated *IRS2* inhibition occurs via the AhR receptor-mediated pathway.

The action of insulin is initiated by binding to its cognate receptor. The activated receptor, in turn, recruits and phosphorylates a panel of substrate molecules. Among these, IRS-1 and IRS-2 appear to be the adapter molecules playing a major role in the coupling to the PI3K-PKB downstream kinases[37]. Tyrosine-phosphorylated IRS-1/2 recruits the heterodimeric p85/p110 PI3K at the plasma membrane where it produces the lipid second messenger PIP3. This action in turn activates the serine/threonine–protein kinase B (PKB)/Akt, Akt regulates the insulin-stimulated translocation of the glucose, resulting in increased glucose uptake. Our data show that H-Exo and PC treatment not only inhibit insulin-mediated induction of IRS-2 but also led to reduced activation of PI3K and Akt phosphorylation that plays an important role in insulin signaling-mediated uptake of glucose (Fig. 6e, f, respectively). IRS-2 is the responsive adaptive molecule for activation of PI3K and Akt[38,39]. These effects were abolished in *AhR$^{-/-}$* mice (Fig. 6f), suggesting AhR crosstalks with the insulin signaling pathway.

Further, the additional evidence for AhR involvement in the insulin -signaling pathway and the downregulation of IRS-2 expression is provided from *AhR$^{-/-}$* mice where AhR was re-expressed via Ad-AhR tail injection. Western blots of liver extracts from these mice suggest that re-expression of AhR led to a reduction in pIRS-2 protein levels (Fig. 6g).

Finally, to confirm if IRS-2 was involved in H-Exo-mediated inhibition of glucose uptake, we overexpressed IRS-2 in hepatocytes, differentiated 3T3 adipocytes, and mouse muscular cells. No inhibition of glucose uptake was observed in IRS-2 overexpressed in hepatocytes, adipocytes, or mouse muscular cells treated with extracts from the liver, adipose tissue, or muscular tissue of mice treated with H-Exo compared with L-Exo or PBS treatment (Fig. 6h, i). This result suggests that *IRS-2* is an essential gene for H-Exo-mediated inhibition of glucose uptake.

We also noticed that unlike *IRS2*, the expressed level of other genes affected by H-Exo in the liver did not have a pattern similar to the pattern in adipose or muscular tissues (see Fig. 6a, b). Whether this difference is associated with the intensity of the H-Exo signal in each organ was further investigated. In vivo imaging results indicated that the intensity of the H-Exo signal (Fig. 6j) is different from the highest intensity in the liver and the lowest in muscle tissue.

The fold change of expressed genes listed in Fig. 6k agreed with the intensity of H-Exo signal, except for *LDLR*, *INS1*, and *UCP1*. Collectively, these results suggest that H-Exo treatment leads to altering the expression of the genes that regulate the insulin-signaling pathway in all metabolite tissues. The H-Exo-mediated altering expression of genes is tissue-dependent.

**H-Exo-mediated activation of AhR leads to inducing dyslipidemia in mice.** Because AhR is known to be involved in cholesterol synthesis[40], high-fat intake induces insulin resistance and dyslipidemia, including high cholesterol and triglycerides[41]. We next assessed alternations in liver steatosis, plasma cholesterol triglycerides, ALT, and AST levels from HFD-fed mice treated with L-Exo or H-Exo for 14 days. H&E staining of liver tissues of H-Exo recipient mice gave the impression of liver steatosis compared to PBS or L-Exo recipient mice (Fig. 7a). H-Exo-treated C57BL/6 and C57BL/6 germ-free mice showed significantly elevated levels of plasma cholesterol and triglycerides, whereas *AhR$^{-/-}$* mice showed no significant change in plasma cholesterol and triglycerides (Fig. 7b). Furthermore, plasma ALT and AST levels were significantly elevated in H-Exo-treated mice vs PBS and L-Exo-treated mice. This result is associated with liver damage (Fig. 7c). Collectively, these results suggest that activation of AhR-mediated signaling contributes to H-Exo-induced dyslipidemia and liver damage.

**Discussion**

In this study, we demonstrated that an HFD affects the composition of intestinal epithelial exosomes, an understudied intestinal

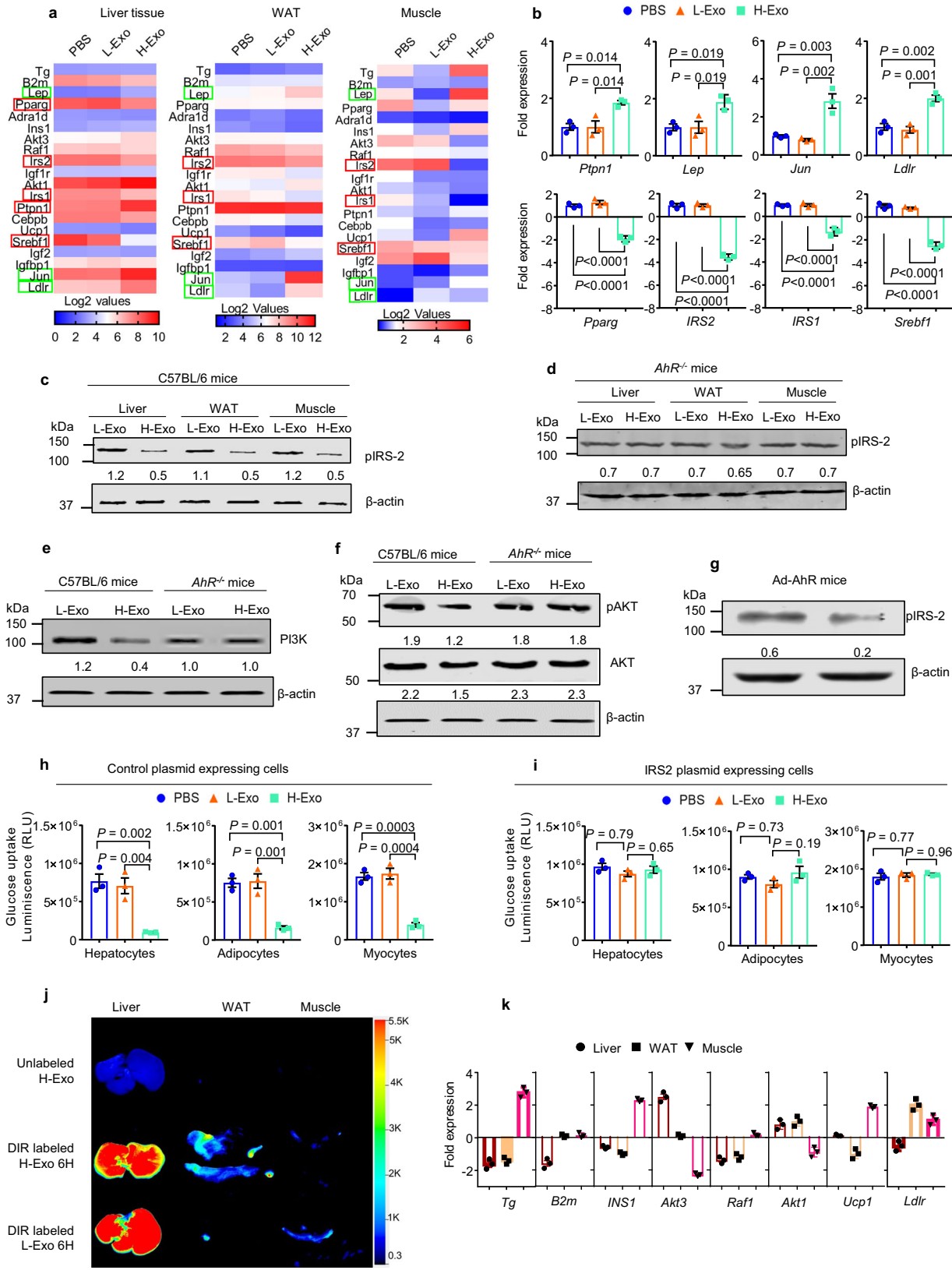

component. Indeed, H-Exo PC plays a role in the development of insulin resistance in both SPF and germ-free mouse models. The effects of H-Exo were abrogated by depleting exosomal PC, suggesting that insulin resistance in humans could be mediated by diet-induced changes to intestinal exosomal lipids.

Overproduction of PC contributes to human metabolic disorders[42]. We found that the diet-dependent increase in exosomal PC is clinically applicable because abnormally high and abnormally low levels of PC in various tissues may influence energy metabolism and have been linked to disease progression[43].

**Fig. 6 H-Exo impact on insulin signaling in liver, adipose, and muscle tissues. a** Gene expression heat map from the insulin-signaling PCR array (Qiagen) performed for liver, adipose, and muscle tissues derived from mice receiving 14 days of oral administration of CD63$^+$A33$^+$ exosomes. **b** Upregulated or downregulated genes in liver tissues derived from mice treated with PBS, L-Exo. and H-Exo ($n = 3$). Filled circle—PBS, filled triangle—L-Exo, and filled rectangle—H-Exo. **c, d** Western blots of pIRS-2 in liver, adipose, and muscle tissue extracts from C57BL/6 (**c**) and AhR$^{-/-}$ mice (**d**). Ratio to β-actin shown in the middle. **e, f** Western blots of PI3K, pAKT and AKT in liver tissue extracts from C57BL/6 and AhR$^{-/-}$ mice. Ratio to β-actin shown in the middle as numbers. **g** AhR was re-expressed in the liver via tail injection of Ad-AhR (adenovirus). Mice were treated with L-Exo or H-Exo for 14 days. pIRS-2 levels measured in liver lysates by western blot. Ratio to β-actin shown in the middle as numbers. **h, i** Glucose uptake measured in tissue extract-treated hepatocytes, adipocytes, and myocytes transfected with either control (**h**) or IRS2 (**i**) plasmid vectors ($n = 3$). **j** Scanning images of liver, WAT, and muscle for DIR-labeled exosomes. **k** Tissue-specific and H-Exo-dependent alteration in the expression of genes regulating insulin signaling. Expressions are shown in comparison with PBS-treated mice ($n = 3$). Circle—liver, rectangle—WAT, and downward triangle—muscle. Data are presented as the mean ± SD. One-way ANOVA with a Tukey post hoc or two-way ANOVA with a Tukey post hoc test. Source data are provided as a Source Data file.

Further research is needed to determine whether a healthy diet has effects on lowering the concentration of exosomal PC in obese/early type 2 diabetes patients to prevent the development of insulin resistance and whether exosomal PC can be used as a diagnostic biomarker for type 2 diabetes and metabolic-related liver disease. In addition, whether the metabolites released from the exosome recipient cells have harmful or beneficial effects systemically, requires further investigation in vitro and in vivo studies. Our finding that higher PC concentrations on exosomes result in more exosomes being taken up by macrophages provides a strategy for engineering exosomes to be targeting delivery vehicles for a variety of molecules.

An enormous diversity of PC molecular species is present in mammalian cells since the constituents of PC can be remodeled to incorporate palmitic acid (PA), for example. PA incorporation into lipids is not determined by the absolute intake of PA, but by the ratio among different types of fatty acids (FA)[44]. Thus, a conclusive role of PC in the context of exosomes needs to be drawn cautiously if the modification of PC is involved in H-Exo-mediated insulin resistance.

PEMT converts PE to PC and is found in the endoplasmic reticulum and in mitochondria-associated membranes. Mice lacking PEMT have lower levels of PC and are protected from HFD-induced obesity and insulin resistance[45–48]. Our results indicate that the increased PC in the intestinal exosomes of mice fed a HFD is associated with an increased level of PEMT. These findings provide a rationale for further studying whether a healthy diet has inhibitory effects on PEMT expression in the intestinal epithelial cells, whereas other unhealthy diets that are high in fats and/or sucrose enhance PEMT expression via AhR activation[6].

The free form of PC has no targeting specificity with reduction of PC functions in a linear dilution gradient manner. In this study, we demonstrated that liver macrophages and hepatocytes take up more of the PC carried by H-Exo when compared to other cell types. Therefore, it is conceivable that the biological effect of free PC on the host might be different from PC carried by exosomes.

The liver has developed exquisite sensing mechanisms to detect signals from the gut community through the gut–liver axis. We propose that AhR is one such biosensor expressed in hepatocytes. Upon liver AhR-mediated detection of gut metabolites, such as exosomes, activated AhR crosstalks with transcriptional factors to rewire liver cell metabolism and reprogram the cellular transcriptome. Depending on what type of ligands bind to AhR, different transcription factors could selectively crosstalk with each other.

We demonstrated that PC carried by H-Exo interacts with AhR in the recipient cells. Our data further indicate that H-Exo regulates the expression of a number of hepatic genes, including IRS-1, IRS-2, peroxisome proliferator-activated receptor-gamma (PPARγ), Srebf1, Lep, Jun, and Ldlr that all play important roles in the insulin-signaling pathway[49,50]. Among these genes, IRS-2 is an essential gene for the downstream insulin-signaling mechanism and genes downstream of IRS-2 include PI3K and akt. Activation of Akt is known to result in increased glucose uptake[39]. Our results show that H-Exo inhibits the expression of IRS-2 and prevents activation of Akt via AhR signaling. Knockout of AhR in hepatocytes leads to abolishing H-Exo-mediated inhibition of expression of IRS-2 and glucose uptake, suggesting that H-Exo-mediated activation of AhR leads to inducing a number of genes including IRS-2 and others listed in Fig. 6b. Collectively, the genes regulated via H-Exo-mediated activation of the AhR pathway contribute to the regulation of insulin signaling. These findings provide the foundation for further studying whether a healthy diet can block H-Exo-mediated inhibition of IRS-2 and cellular mechanisms underlying how the gut cross-talks with not only liver but other tissues such as adipose and muscular tissue via intestinal exosomes.

H-Exo PC-induced AhR signaling activation contributes to the development of insulin resistance, but constitutive expression of AhR in AhR transgenic mice has been shown to improve insulin sensitivity, despite exacerbating hepatic steatosis by induction of FGF21[51]. AhR is a ligand-activated transcription factor that is activated by small molecules provided by the diet, microorganisms, metabolism, and pollutants. A number of studies have demonstrated that control of AhR functions is challenged by the fact that AhR is often involved in balancing opposing processes. Therefore, depending on the nature of the factors that activate AhR, different panels of genes with different functions could be induced upon AhR activation.

Although in this study we focused on the role of how H-Exo PC in an HFD induces insulin resistance, our finding that a HFD also altered the composition of exosomes provides further opportunities for further studying whether the altered exosomal lipids, proteins, and RNA profile due to HFD feeding contribute to obesity-related metabolic syndromes, including hypertension, heart attack, and stroke. Further, which factor(s) derived from HFD leads to altering the composition of H-Exo needs to be further investigated.

Our data also suggest that the exosomes released from intestinal epithelial cells predominately trafficked to the liver and small amounts also trafficked to adipose and muscle tissue. However, tumor cell-derived exosomes favorably target to immune organs for possible immune modulation such as immune suppression, which is one of the well-known mechanisms underlying tumor immune evasion. In contrast, our findings suggest that the exosomes released from intestinal epithelial cells of HFD-fed mice dysregulate the gut/liver/adipose/muscular tissue axis communication, which is one of the major mechanisms underlying metabolic syndrome. This study provides a foundation for further identifying the molecular mechanism underlying tumor exosomes being favorable to targeting immune organs, whereas non-tumor exosomes target metabolic organs such as liver.

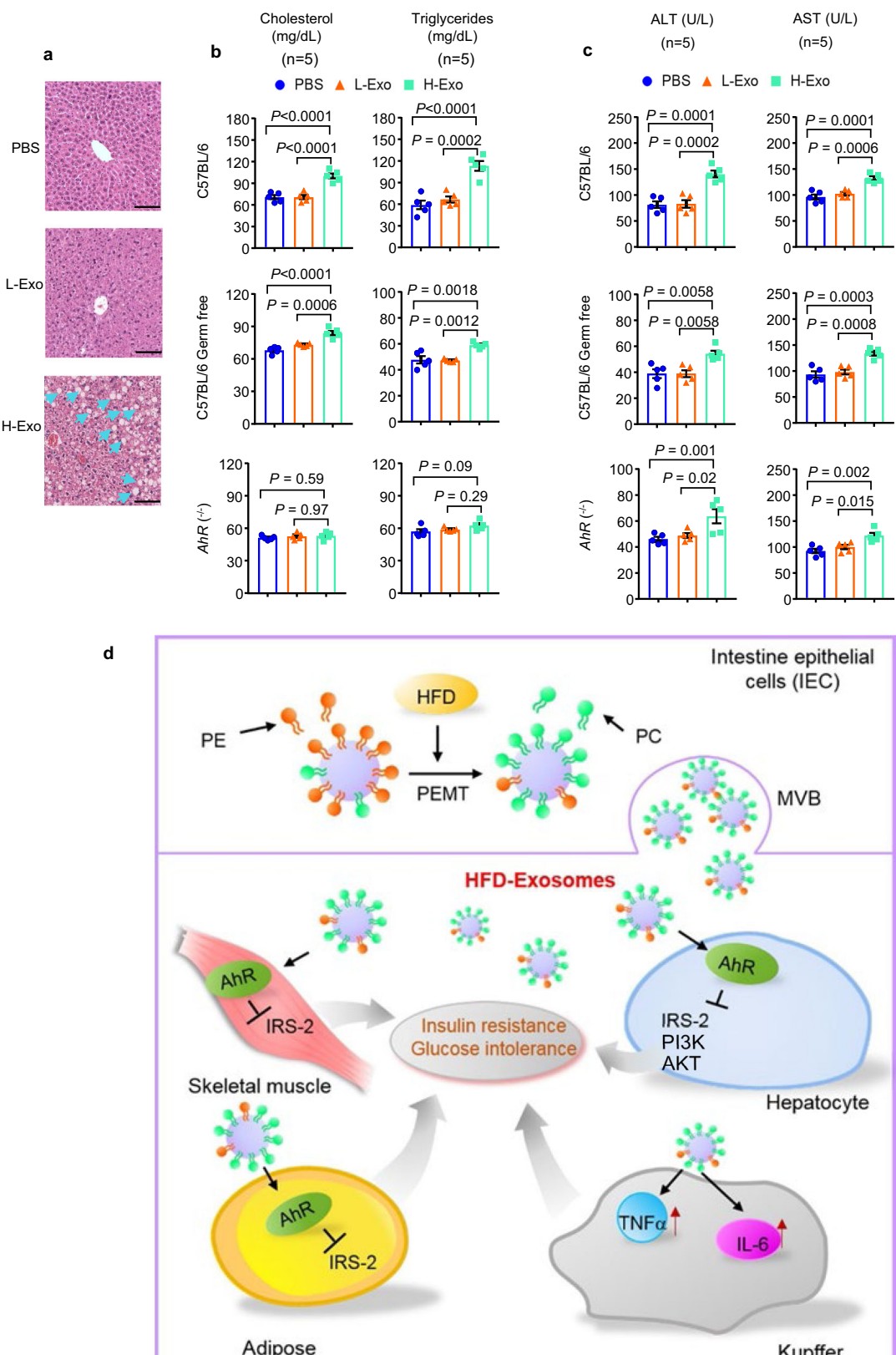

**Fig. 7 H-Exo induced liver steatosis and dyslipidemia in C57BL/6 and C57BL/6 germ-free mice, but not in *AhR*<sup>−/−</sup> mice. a** H&E staining of liver tissue sections of mice receiving PBS, L-Exo, or H-Exo for 14 days. Blue arrows indicate steatosis. The scale bar is 20 μm. Data were repeated at least three times. **b** Plasma cholesterol and triglyceride levels after exosome treatment for 14 days (*n* = 5). Filled circle—PBS, filled triangle—L-Exo, and filled rectangle—H-Exo. **c** Plasma ALT and AST levels (*n* = 5). **d** Model: The intestinal epithelial cells of HFD-fed mice-release CD63⁺A33⁺ exosomes that induce insulin resistance and glucose intolerance via an AhR-mediated pathway. Data are presented as the mean ± SD. One-way ANOVA with a Tukey post hoc test. Source data are provided as a Source Data file.

We also observed that circulating exosomal PC is also increased in HFD-fed mice. This finding provides a rationale for further investigating whether H-Exo PC on circulating exosomes from HFD- fed mice may also have a role in the development of insulin resistance. However, the circulating exosomes are a mixed population that could be released from cells that are resident in a variety of tissues, including intestinal epithelium, adipose, liver, muscles, and many other tissues. Isolating and having enough subsets of cell-specific exosomes from circulating and mixed exosomes for studies of this nature is technically challenging. In contrast to blood exosomes, large amounts of fecal exosomes are released from intestinal epithelial cells. Fecal exosomes are one of major subpopulations of the gut community released daily (1 g of feces/20 g of BW of mouse can be collected per day, $4 \times 10^{10}$ CD63+A33+ exosomes/g feces). Another advantage is that fewer tissues release exosomes into the intestinal lumen; thus, the fecal exosomes are a fairly pure population of exosomes. An additional advantage of fecal exosomes over blood exosomes is that a relatively large quantity of fecal nanoparticles can be isolated based on being CD63 positive[52,53] or negative for CD63 (CD63−A33+) using CD81 and CD9 markers that allow for studying the role of these subsets of exosomes in regulating insulin signaling using the same approach as described in this study.

The large quantity of fecal exosomes also provides enough material for studying a number of challenging questions in nanovesicle trafficking from the digestive system to the circulation. Absorption of either endogenous, such as intestinal epithelial exosomes, or exogenous nanovesicles, such as edible plant exosome-like nanoparticles[54], is a process whereby nanovesicles are transferred from the lumen of the GI tract to the underlying epithelial cells and then further transported to the systemic circulation and various tissues through either the portal vein or the lymphatic system. There are several potential uptake pathways that the indigestible nanovesicles might undergo when passing through the epithelium and entering the systematic circulation, i.e., paracellular transport, transcellular transport, and persorption. However, currently, the mechanism unraveling each specific uptake pathway(s) of intact nanovesicles in vivo is still not clear even though many studies have been conducted using different in vitro cellular models. It is conceivable that the intact nanovesicles may be absorbed at the epithelial lining only if they can successfully permeate through the mucus that covers the epithelium. Mucin molecules create a filter that can block particulates that are too large from permeating the mucus mesh spacing (~200 nm)[55–57]. Thus, particle size plays a central role in the penetration of intact nanovesicles through the mucus layer. A subset of intestinal exosomes with less than 200 nm in size is expected to enter the blood circulation. In addition, due to the negative charge of glycosylated groups in mucins[58]. The mucus can tightly bind nanoparticles through hydrophobic and electrostatic interactions. Therefore, hydrophilicity and the surface-positive charge of nanovesicles are unlikely across the intestine epithelium. Intestinal epithelial exosomes are negatively charged, which may be advantageous so that sufficient numbers pass through the mucus layer. Once the intact nanovesicles have penetrated through the mucus layer, they may then be transported through the epithelium cells mainly via three mechanisms: paracellular transport (i.e., between adjacent epithelial cells), transcellular transport (i.e., through an epithelial cell), and persorption (i.e., through dead or dying cells). The results generated from this study provide a rationale for further determining which mechanism(s) underlies intestinal epithelial cell exosomes trafficking from the GI tract to the bloodstream.

In this study, C57BL/6 mice administered H-Exo for 14 days developed insulin resistance while being fed the HFD. The HFD feeding seems to accelerate the developing insulin resistance

compared with the RCD-fed mice that developed insulin resistance after a 28-day feeding (Supplementary Fig. 17).

Overall, our study as summarized in our model (Fig. 7d) reveals a different role for intestinal exosomes in mediating communication of the gut and three major metabolic organs. Diet alters the composition and therefore the function of intestinal exosomes. Insulin resistance is induced in an exosomal PC-dependent manner by activation of the AhR receptor-mediated pathway. The results lend credence to future research examining whether gut community-derived factors that can prevent PC recruitment into intestinal exosomes may be useful in the prevention or treatment of type 2 diabetes in humans.

## Methods

**Mice**. In total, 8- to 12-week-old male C57BL/6 mice were purchased from the Jackson Laboratory (Bar Harbor, ME) and maintained on a 12-h/12-h light/dark cycle in a pathogen-free animal facility at the University of Louisville. Mice were fed a RCD or an HFD (60% Fat, Catalogue, D12492, Research Diet Inc.) during the study. AhR-knockout mice were purchased from Taconic Biosciences (Rensselaer, NY). Germ-free mice were purchased from the National Gnotobiotic Rodent Resource Center (University of North Carolina, Chapel Hill, NC) and maintained in flexible film isolators (Taconic Farm) at the Clean Mouse Facility of the University of Louisville. Animal care was performed following the Institute for Laboratory Animal Research (ILAR) guidelines, and all animal experiments were conducted in accordance with protocols approved by the University of Louisville Institutional Animal Care and Use Committee (Louisville, KY).

**Antibiotic treatment of mice**. An antibiotic mixture of ampicillin, metronidazole, neomycin, streptomycin, each at 1 mg/ml, and vancomycin at 0.5 mg/ml were added to drinking water. Mice were treated with the antibiotic mixture for 7 days[59]. Feces from mice receiving the antibiotic mixture were collected to isolate the exosomes. Purified exosomes were tested for endotoxin presence and endotoxin-free exosomes were used for oral administration to germ-free mice. Germ-free mice were maintained in the germ-free facility of University of Louisville, Louisville, KY, USA.

**Cells**. Murine hepatocytes (FL83B) and human hepatocytes (HepG2) (American Type Culture Collection, ATCC) were grown in tissue culture plates with F12K medium (Thermo Fisher Sci) supplemented with 10% heat-inactivated fetal bovine serum (FBS), 100 U ml− penicillin, and 100 mg ml−1 streptomycin at 37 °C in a 5% $CO_2$ atmosphere. Human monocytes (U937) were grown in RPMI 1640 medium (Thermo Fisher Sci) supplemented with 10% FBS. For fecal exosomes (with or without PKH26 labeling) treatment for FL83B and HepG2 cells, $2 \times 10^4$ cells were seeded into six-well plates. After achieving 50–60% confluence, fecal exosomes (numbers indicated in figures) were added and incubated for 16 h at 37 °C in a 5% $CO_2$ atmosphere. Cells were washed with PBS and processed for imaging, RNA isolation, and protein extraction. The C57BL/6 murine colon epithelial MC-38 cell line and human embryonic kidney 293 cells (ATCC) were grown at 37 °C in 5% $CO_2$ in Dulbecco's modified Eagle's medium (DMEM, Thermo Fisher Sci.) supplemented with 10% heat-inactivated FBS, 100 U ml−1 penicillin, and 100 μg ml−1 streptomycin.

Mouse-immortalized 3T3-L1 fibroblasts were grown at 37 °C in 5% $CO_2$ in Dulbecco's modified Eagle's medium (DMEM, Thermo Fisher Sci.) supplemented with 10% heat-inactivated FBS, 100 U ml−1 penicillin, and 100 μg ml−1 streptomycin, Life Technologies. Cells were induced to differentiate for 2 days after reaching confluence (day 0) by supplementing growth media with 3 nM insulin (Humulin R, Eli Lilly), 0.25 μM dexamethasone (Sigma-Aldrich), and 0.5 mM 1-methyl-3-isobutyl-xanthine (Sigma-Aldrich). From day 3 until day 7, cells were maintained in growth media supplemented with 3 nM insulin after which the mature adipocytes were maintained in growth media.

The C2C12 skeletal muscle cell line was grown at 37 °C in 5% $CO_2$ in DMEM supplemented with 10% heat-inactivated FBS, 100 U ml−1 penicillin, and 100 μg ml−1 streptomycin. Myogenic differentiation was started upon reaching 40% confluence by switching the cells to a medium containing 2% horse serum supplemented with insulin (10 nm). For the glucose uptake assay, both cell types were cultured with PBS, L-Exo, or H-Exo for 12−16 h at 37 °C in a 5% $CO_2$ atmosphere.

**Human subjects**. The study involved five healthy volunteers between the ages of 25 and 45 years old (all males) and seven patients with type 2 diabetes (T2D; characteristics mentioned in Table 1). No healthy volunteers had a history of chronic gastrointestinal disease. All volunteers were recruited from the University of Louisville Hospital, Louisville, Kentucky, USA. Type 2 diabetes was diagnosed according to the American Diabetes Association diagnostic criteria (American Diabetes Association 2012). All clinical fecal samples were collected from patients in the outpatient endocrinology clinic. The study was conducted in accordance to the criteria set by the declaration of Helsinki. All participants were educated

**Table 1 Details of human individuals involved in the study.**

|  | Healthy subjects, $n = 5$ (mean ± S.D) | T2D patients, $n = 7$ (mean ± S.D) |
|---|---|---|
| Age (years) | 50.2 ± 7.75 | 56 ± 12.5 |
| BMI | 21.8 ± 1.25 | 31.8 ± 3.46 |
| HOMA-IR | 2.24 ± 0.15 | 7 ± 0.65 |
| HbA1c % | 4.68 ± 0.32 | 8.2 ± 0.37 |
| T2D duration (years) | No chronic disease | 10.5 ± 3.15 |
| Total cholesterol (mg/dL) | 149.2 ± 8.87 | 139 ± 9.04 |
| Triglycerides (mg/dL) | 138.4 ± 7.50 | 189 ± 11.7 |

regarding their participation and signed a written consent form. Approval for the study was granted by the University of Louisville Research Ethics Committee.

**Isolation and purification of fecal exosomes**. Fecal pellets (pooled from 10 mice) were resuspended in phosphate-buffered saline (PBS) and minced manually. Differential centrifugation was deployed to isolate the fecal exosomes. Fecal suspensions were centrifuged at 1000$g$ for 10 min, 2000$g$ for 20 min, and 4000$g$ for 30 min to remove large particles. The supernatant was centrifuged at 8000$g$ for 1 h to remove microparticles. Finally, each suspension was centrifuged at 100,000$g$ for 2 h. Pellets were suspended in PBS. The exosomes were further purified by sucrose gradient (8, 30, 45, and 60% sucrose in 20 mM Tris–Cl, pH 7.2) centrifugation. An aliquot of the purified exosomes was fixed in 2% paraformaldehyde for transmission electron microscopy (EM) using a conventional procedure and observed using an FEI Tecnai F20 (EM facility at the University of Alabama, Alabama, USA)[54,60,61]. The EM was done with the following settings: 80 kV at a magnification of 15,000 and defocus of 100 and 500 nm.

**Fecal supernate preparation**. Fresh feces beads were used for the preparation of fecal metabolites. Beads were resuspended in 1× PBS and gently minced with the piston. The suspension was centrifuged at 1000$g$ for 5 min to remove large debris. The supernatant was collected into fresh tubes and centrifuged at 6000$g$ for 30 min to remove microbes[62]. The supernates from this step were used to stimulate the cells transfected with luciferase plasmids.

**Luciferase assay**. pGL3B-PEMT-luc plasmid-transfected MC-38 cells were treated with 100 μl of fecal metabolites from either RCD or HFD mice for 16 h at 37 °C in a $CO_2$ incubator. Luciferase activity was measured using a dual-luciferase system (cat. No. E1910) from Promega Corp. WI, USA, as per the manufacturer's instructions.

**Endotoxin detection in fecal exosomes**. Endotoxin in fecal exosomes was detected using a Pierce$^{TM}$ Chromogenic Endotoxin Quant kit (Cat. No. A39552S). In brief, all reagents were prepared according to the manufacturer's instructions. About 50 μl of endotoxin standard or test samples were added per well (triplicates) of the plate. About 50 μl of amebocyte lysate reagent was added into each well, mixed, and the plate incubated for 30 min at 37 °C. About 100 μl of the prewarmed chromogenic substrate was added into each well and vigorously mixed and the plate incubated for 6 min at 37 °C. About 50 μl of stop solution (25% acetic acid) was added and mixed. The plate was read for optical density at 405 nm. A standard graph was plotted and calculations for endotoxin in test samples done accordingly.

**Nanoparticle tracking analysis**. Purified exosome samples were analyzed for particle concentration and size distribution using the nanoparticle tracking analysis method provided using the Malvern NanoSight NS300 (Malvern Instruments Ltd., Malvern, United Kingdom)[61,63]. The assays were performed in accordance with the manufacturer's instructions. Briefly, for the NanoSight, three independent replicates of diluted exosome preparations in PBS were injected at a constant rate into the tracking chamber using the provided syringe pump. The specimens were tracked at room temperature (RT) for 60 s. Shutter and gain were manually adjusted for optimal detection and were kept at optimized settings for all samples. The data were captured and analyzed with NTA Build 127 software (version 2.2, Malvern Instruments Ltd., Malvern, UK).

For labeled or stained exosomes, the sample was first run without any fluorescent channel and then the sample was run into a specific (PE) fluorescent channel. Percent positivity was calculated as fluorescent-positive exosomes/total exosomes X (100).

**Immunoisolation of exosomes**. The method for immunoisolation of exosomes described elsewhere was followed[64]. Briefly, antibodies for immunoisolation (mouse monoclonal anti-human CD63 (NBP2-32830 0.1 mg, Novus Biologicals) and normal mouse polyclonal IgG (Millipore, Cat. No. 12-371) at a ratio of 1 μg of

antibody per 100 μL of beads were coupled to Pierce Protein A Magnetic Beads (Dyna beads) by incubation at 4 °C overnight. Beads were then washed three times with 500 μL of PBS 0.001% Tween, resuspended in 500 μL of the same buffer, to which exosomes ($2 \times 10^{10}$) were added, followed by overnight incubation at 4 °C with rotation. Bead-bound exosomes were collected and washed three times in 500 μL of PBS-Tween. Exosomes were eluted with high-salt buffer and washed again and centrifuged at 100,000$g$ for 1 h at 4 °C in a TLA 110 rotor (Beckman, Optima TL100 centrifuge).

**Quantitative reverse transcription PCR analysis for mRNA expression**. Total RNA was isolated from tissue and cells using a miRNeasy mini kit (Qiagen). For analysis of AhR, IRS-2, Cyp1a1, Cyp1a2, Cyp1b1, IGF1R, IGF2, LDLR, PTPRF, and JUN mRNA expression, 1 μg of total RNA was reverse transcribed using Super-Script III reverse transcriptase (Invitrogen), and quantitation was performed using primers (Eurofins) with QuantiTect SYBR Green PCR (Qaigen). GAPDH was used for normalization. The primer sequences are listed in Supplemetary Table 5. qPCR was run using the BioRad CFX96 qPCR System with each reaction run in triplicate. Analysis and fold changes were determined using the comparative threshold cycle (Ct) method[60,61]. The change in miRNA or mRNA expression was calculated as fold change.

**miRNA PCR microarray**. Total RNA containing small RNA was isolated from tissue and cells using a miRNeasy mini kit (Qiagen, cat. no. 217004). miRNA expression profiling for exosomes was performed using the Qiagen miScript miRNA PCR Array Mouse miRBase Profiler (Cat# 331223) and an Applied Biosystems ViiA 7 Real-Time PCR System as described[65]. Normalization to endogenous control genes included SNORD61, SNORD68, SNORD72, SNORD9, and RNU6 to correct for potential RNA input or RT efficiency biases. miRNA data generated from exosomes were comparatively analyzed by the online free data analysis software at https://dataanalysis.qiagen.com. Quantile normalization and subsequent data processing were performed and scatterplots representing differentially regulated miRNAs were generated.

**Immunostaining of exosomes**. Immunostaining was carried out using a method described previously[66] with some modifications. Exosomes suspended in PBS were incubated with 5% bovine serum albumin (BSA) for 1 h at RT and washed three times with PBS and primary antibodies added at (1:1000) or directly conjugated antibodies and incubated at 4 °C overnight. The mixture was washed three times with PBS and fluorescent secondary antibody (1:2000 dilution) was added and incubated at RT for 1 h. The mixture was washed again with PBS and centrifuged. The pellet was dissolved in PBS. Finally, the pellet was resuspended into PBS and passed through a 200 nM syringe filter to break up clumps or aggregated exosomes.

**LC–MS analysis**. LC–MS of exosomal proteins was carried out using a method described previously[60,61]. Proteome Discoverer v1.4.1.114 (Thermo) was used to analyze the data collected by the mass spectrometer. The database used in Mascot v2.5.1 and SequestHT searches and the version of the Mus musculus (strain C57BL/6) proteome from Uniprot KB (Proteome ID UP000000589). In order to estimate the false discovery rate, a Target Decoy PSM Validator node was included in the Proteome Discoverer workflow. MS2 scan data from the Xcalibur RAW file were parsed for separate searches of CID and ETD MS2 scans in Mascot and Sequest, and collection of the results into a single file (.msf extension). The resulting.msf files from Proteome Discoverer were loaded into Scaffold Q + S v4.4.5 (Proteome Software, Portland, OR, USA). The scaffold was used to calculate the false discovery rate using the Peptide and Protein Prophet algorithms. Proteins were grouped to satisfy the parsimony principle.

**Lipid extraction from fecal and plasma exosomes and A33$^+$ cells or tissues**. Total lipids were extracted from a sucrose gradient band of processed fecal exosomes. Briefly, 1.9 ml of a 2:1 (v/v) MeOH:CHCl$_3$ mixture was added to 0.5 ml ($2 \times 10^{12}$; in case of cells, two million FACS-sorted cells were used for lipid extraction) of exosomes in PBS. About 0.625 ml of CHCl$_3$ and water (1:1) were added sequentially and vortexed thoroughly. The aqueous and organic phases were separated by centrifugation at 850$g$ for 10 min at 22 °C in glass tubes. The organic phase was collected using a glass pipette. The organic phase was aspirated and dispensed into fresh glass tubes. The organic phase was dried by heating under nitrogen (2 psi) and dried overnight under vacuum. Total lipids were determined using the phosphate assay as described[54,67].

**Lipidomic analysis with triple-quadrupole MS**. Lipids extracted from fecal exosomes were submitted to the Lipidomics Research Center, Kansas State University (Manhattan, KS) for analysis using MS[54,61,68]. In brief, the lipid composition was determined using triple-quadrupole MS (Applied Biosystems Q-TRAP, Applied Biosystems, Foster City, CA). The data are reported as the concentration (nmol mg$^{-1}$ fecal exosomes) and percentage of each lipid within the total signal for the molecular species determined after normalization of the signals to internal

standards of the same lipid class. Multiples correlation analysis was used to predict changes in the lipid content and which lipids were most affected.

*Lipid data processing and analysis*. An automated electrospray ionization–tandem MS approach was used. In this approach, plasma lipid species are identified at the level of head group plus total acyl carbons: total double bonds. The detected intensities were each defined by an intact ion mass/charge (m/z) and a characteristic fragment m/z. Data acquisition and analysis were carried out as described previously[69,70] with modifications. The lipid extracts were introduced by continuous infusion at 30 µl/min into the ESI source on a triple-quadrupole MS/MS (4000QTrap (Applied Biosystems)) or Xevo TQS (Waters Corporation) with an autosampler using a loop. Precise amounts of internal standards, obtained and quantified as previously described[71], were added in the following quantities: 0.60 nmol PC(12:0/12:0), 0.60 nmol PC(24:1/24:1), 0.60 nmol LPC(13:0), 0.60 nmol LPC(19:0), 0.30 nmol PE(12:0/12:0), 0.30 nmol PE(23:0/23:0), 0.30 nmol LPE(14:0), 0.30 nmol LPE(18:0), 0.30 nmol LPG(14:0), 0.30 nmol LPG(18:0), 0.30 nmol PA (14:0/14:0), 0.30 nmol PA (phytanoyl/phytanoyl), i.e., PA(20:0/20:0), 0.20 nmol PS (14:0/14:0), 0.20 nmol PS(phytanoyl/phytanoyl), i.e., PS(20:0/20:0), 0.23 nmol PI (16:0/18:0), 2.5 nmol CE(13:0), and 2.5 nmol CE(23:0). The sample and internal standard mixture were combined with solvents, such that the ratio of chloroform/methanol/300 mM ammonium acetate in water was 300/665/35, and the final volume was 1.2 ml. This mixture, in autosampler vials, was centrifuged for 15 min to pellet particulates before presenting the lipid/solvent mixture to the autosampler. These unfractionated lipid extracts were introduced by continuous infusion into the ESI source on a triple-quadrupole MS/MS (API 4000, Applied Biosystems, Foster City, CA), using an autosampler (LC Mini PAL, CTC Analytics AG, Zwingen, Switzerland) fitted with the required injection loop for the acquisition time and presented to the ESI needle at 30 ml/min. Sequential precursor and neutral loss scans of the extracts produced a series of spectra with each spectrum revealing a set of lipid species containing a common head group fragment.

Lipid species were detected with the following scans: PC, SM, and lysoPC, [M + H] + ions in positive ion mode with Precursor of 184.1 (Pre 184.1); PE and lysoPE, [M + H] + ions in positive ion mode with Neutral loss of 141.0 (NL 141.0); PI, [M + NH4] + in positive ion mode with NL 277.0; PS, [M + H] + in positive ion mode with NL 185.0; PA, [M + NH4] + in positive ion mode with NL 115.0; CE, [M + NH4] + in positive ion mode with Pre 369.3. SM was determined from the same mass spectrum as PC (precursors of m/z 184 in positive mode)[71] and by comparison with PC internal standards. For each spectrum, 9–150 continuum scans were averaged in multiple channel analyzer (MCA) mode. The source temperature (heated nebulizer) was 100 µC, the interface heater was on, +5.5 kV or 24.5 kV were applied to the electrospray capillary, the curtain gas was set at 20 (arbitrary units), and the two ion source gases were set at 45 (arbitrary units). The background of each spectrum was subtracted, the data were smoothed, and peak areas integrated using a custom script and Applied Biosystems Analyst software. The data were isotopically deconvoluted, and the lipids in each class were quantified (molar amounts) in comparison to the internal standards of that class.

For TAG analyses, the background of each spectrum was subtracted, the data were smoothed, and peak areas were integrated using a custom script and Applied Biosystems Analyst software. Peaks corresponding to the target lipids in these spectra were identified, and the data were corrected for A + 2 isotopic overlap (based on the mass-to-charge ratio, m/z, of the charged fragments) within each spectrum. Signals were also corrected for isotopic overlap across spectra, based on the A + 2 overlaps and masses of the neutral fragments. A sample containing internal standard alone, run through the same series of scans, was used to correct for chemical or instrumental noise: amounts of each target lipid detected in the "internal standard-only" sample were subtracted from the molar amounts of each target lipid calculated from the plant lipid spectra. The 'internal standard-only' spectra were used to correct the data from the following five samples run on the instrument. The extracted data from all acyl NL scans at each TAG mass, as defined by m/z, which corresponds to total acyl carbons: total acyl double bonds, were used to calculate the amount of each individual TAG species.

**HPLC analysis of PE and PC**. The lipids extracted from fecal exosomes were diluted with an equal volume of methanol and filtered through a 0.22 nM filter. In all, 25 µl of lipids in methanol were injected for HPLC analysis. The HPLC analysis was performed on an Agilent 1260 Infinity system equipped with an Agilent 300, SB-C8 column (4.6 × 250 mm, 5 µm), with the following parameters: mobile phase A: water with 0.1% formic acid; mobile phase B: 100% acetonitrile modified with 0.1% formic acid (v/v); gradient: 10% B in first 5 min, 10–95% B for 10 min, hold 95% B for 5 min, 95%–10% B for 5 min, with a 2 min post run. Flow rate: 0.5 ml/min; temperature: 30 °C. UV detection at 220 nm was used to monitor PE and PC. The standard for PE (cat: 841118C-25mg) and PC (cat: 850458C-25mg) were purchased from Avanti Polar Inc. (USA). The standard stock solutions were dissolved in chloroform/methanol 1:1 (v/v) at a concentration of 1 mM and stored at −20 °C. Lipid standard mixtures were prepared fresh every day in chloroform/methanol 1:1 (v/v) at a concentration of 3 µM and used immediately.

**Transfection experiments**. Mouse hepatocytes (FL83B) were transfected with 200 ng of constructs pGL3B-PEMT-Luc (kindly provided by Dr. Jongsook Kim Kemper, Department of Molecular and Integrative Physiology, the University of

Illinois at Urbana, IL, USA) and pBABE puro mouse IRS-2 (cat. No. 11371, purchased from addgene Watertown, MA, USA) were used. Transfections were performed using kit from Invitrogen (cat. No. L3000-015) in accordance with the manufacturer's instructions.

**Glucose and insulin tolerance tests (GTT and ITT)**. In total, $2 \times 10^9$/dose/day exosomes were administered orally to each mouse for 14 days. For glucose tolerance tests, after an overnight fast, baseline glucose levels were determined using the glucometer (Priology, USA). Mice were then given an intraperitoneal injection of glucose (dextrose) at a dose of 2 mg g$^{-1}$ of body weight[72]. The blood glucose levels were measured at 30, 60, 90, and 120 min after glucose injection. For insulin tolerance tests, mice were fasted for 4–6 h and basal blood glucose levels were determined. Then mice were given an intraperitoneal injection of insulin (1.2 units g$^{-1}$ of body weight). The blood glucose levels were measured at 30, 60, and 90 min (otherwise indicated in figures) after insulin injection.

**Insulin quantification in plasma**. Plasma insulin was quantified using a Mouse Insulin ELISA kit (EMINS, ThermoFisher Scientific) in accordance with the manufacturer's instructions. In brief, insulin standards were prepared by serial dilutions (1:1). All plasma samples were diluted 2-fold with the provided diluent buffer. In all, 100 µl of each standard and sample were added into appropriate wells. Wells were covered and incubated for 2.5 h at RT. Wells were washed four times with wash buffer and 100 µl of 1× biotinylated anti-insulin antibody added to each well and incubated for 1 h at RT with gentle shaking. After four washes, 100 µl of streptavidin–horseradish peroxidase (HRP) solution was added to each well and incubated for 45 min at RT with gentle shaking. Again, the wash steps were repeated, and 100 µl of TMB solution was added to each well and incubated for 30 min at RT in the dark with gentle shaking. Absorbance was measured at 450 and 550 nm within 30 min after adding the stop solution.

**Adoptive transfer of exosomes to mice**. A single dose of $2 \times 10^9$ exosomes (L-Exo or H-Exo) in 200 µl of PBS were orally administered on a daily basis for 14 days while mice were fed HFD (diet composition mentioned in Supplementary Table 4). In total, mice received 14 doses of $2 \times 10^9$ exosomes in 14 days. Mice were monitored for any changes or effects and were kept at the University of Louisville animal facility. After 14 days of adoptive transfer, the weight of mice was taken, and GTT and ITT tests were performed as described in the previous section. In case of adoptive transfer of human exosomes to mice, exosomes from seven T2D patients were pooled for T2D-Exo and five healthy subjects were pooled for the Healthy-Exo group.

**Metabolic monitoring and nuclear magnetic resonance for body composition**. Bodyweight and body composition were measured in ad-lib fed conscious mice using an EchoMRI™ 3-in-1 system nuclear magnetic resonance spectrometer (Echo Medical Systems, Houston, TX) to determine whole-body lean and fat mass. Mice were individually placed in the Oxymax Lab Animal Monitoring System (Columbus Instruments) with free access to food and water. After overnight acclimation, oxygen consumption, the amount of carbon dioxide produced, food and liquid intake, and locomotor activity (beam breaks) were determined. Respiratory exchange ratios were calculated by dividing the volume of carbon dioxide produced by the volume of oxygen consumed ($VCO_2/VO_2$).

**Hyperinsulinemic–euglycemic clamps**. Clamp studies were performed in mice at the Penn Diabetes Research Center Rodent Metabolic Phenotyping Core (University of Pennsylvania). Indwelling jugular vein and carotid artery catheters were surgically implanted in the mice for infusion 7 days prior to the clamp study day as previously described[73]. Mice were fasted for 5 h prior to initiation of the clamp and acclimated to the containers (plastic bowl with alpha dry). The jugular vein and arterial lines were connected to the dual swivel 2 h prior to the clamp initiation. A [3-$^3$H] glucose infusion was primed (1.5 µCi) and continuously infused for a 90-min equilibration period (0.075 µCi/min). Baseline measurements were determined in arterial blood samples collected at −10 and 0 min (relative to the start of the clamp) for analysis of glucose, [3-$^3$H] glucose-specific activity, basal insulin, and FFA. The clamp was started at $t = 0$ min with a primed continuous infusion of human insulin (2.5 mU/kg/min, Novolin Regular Insulin), a donor blood infusion at 4.5 µl/min to prevent a 5% drop in the hematocrit, and glucose (D50 mixed with [3-$^3$H]glucose 0.05 µCi/µl) was infused at variable GIR to maintain euglycemia. The mixing of D50 with [3-$^3$H] glucose is required to maintain the specific activity constant during the clamp period. Arterial blood samples were taken at $t = 80$–120 min for the measurement of [3-$^3$H] glucose specific activity, clamped insulin, and FFA levels. In all, 120 min after initiation of clamp, 14C-2DG (12µCi) was injected and arterial blood samples obtained at 2, 5, 15, 25, and 35 min to determine Rg, an index of tissue-specific glucose uptake in various tissues. After the final blood sample, animals were injected with a bolus of pentobarbital, and tissues were collected and frozen in liquid nitrogen and stored at −80 °C for subsequent analysis.

**Processing of samples and calculations**. Radioactivity of $[3-^3H]$glucose, $[^{14}C]$ 2DG, and $[^{14}C]$2DG-6-phosphate were determined as previously described[74]. The glucose turnover rate (total $Ra$; mg/kg/min) was calculated as the rate of tracer infusion (dpm/min) divided by the corrected plasma glucose specific activity (dpm/ mg) per kg body weight of the mouse. Glucose appearance ($R_a$) and disappearance ($R_d$) rates were determined using steady-state equations and endogenous glucose production ($R_a$) was determined by subtracting the GIR from total $R_a$. Tissue specific glucose disposal (Rg; μmol/100 g tissue/min) was calculated as previously described[74]. The plasma insulin concentration was determined using a mouse ELISA kit (Biomarker core, University of Pennsylvania).

**Isolation and culture of primary hepatocytes**. Totally, 8–10-week-old C57/Bl6 male mice were anesthetized by injecting a mixture of Ketamine (80 mg/kg) and Xylazine (5 mg/kg) in 200 μL of saline intraperitoneally. The liver was perfused with pre-warm Solution 1 Hank's Balanced Salt Solution (HBSS, Gibco), EDTA 0.5 mM, pH = 8, solution 2 Dulbecco's Modified Eagle's Medium (DMEM, Gibco), Collagenase Type I (0.8 mg/mL). Each liver (severed the portal vein and the connective tissue and separate from the body) was collected into a tube containing 15 mL of Solution 2. The liver sac was cut to release the hepatocytes. Totally, 35 mL of Solution 3 (Dulbecco's Modified Eagle's Medium) was added and the liver suspension was passed through a cell strainer (using a spatula to gentle stir tissue on the strainer). The cells were spun at 50 g for 1 min and the pellet washed twice with Solution 3 (45 g, 1 min). The supernatant appeared clear after the last wash. Cell culture plates were coated for 24 h with 0.1% fibrinogen (Sigma) before plating cells. After isolation, cell number and viability were determined. The cell pellet was re-suspended by gently pipetting cells up and down in the appropriate volume of plating medium (at 37 °C). Cells were seeded gently into the culture plate and incubated in a tissue culture incubator set at 37 °C, 5% $CO_2$. Once the cells were attached, the cells were treated with PBS, L-Exo, or H-Exo for 16 h[54,75].

**TLC analysis**. Total lipids from fecal exosomes were quantitatively analyzed using a method previously described[54] and used for TLC analysis[67]. Briefly, HPTLC-plates (silica gel 60 with a concentrating zone, 20 cm × 10 cm; Merck) were used for the separation. After extracting samples of concentrated lipid from fecal exosomes, the lipids were separated on a plate that had been developed with chloroform/ methanol/acetic acid (190:9:1, by vol). After drying in air, the plates were sprayed with a 10% copper sulfate and 8% phosphoric acid solution and then charred by heating at 120 °C for 5 - 10 min. The bands of lipid on the plate were imaged using an Odyssey Scanner (Licor Bioscience, Lincoln NE).

**Generation of the GFP-MC38 cell line releasing green fluorescent protein (GFP) exosomes**. A lentivirus preparation was made using a previously described method[60]. The PalMGFP expressing lentivirus was generated by transfecting the GFP expression plasmids (PalMGFP; kindly provided by Xandra O. Breakefield, Department of Neurology and Radiology, Massachusetts General Hospital, Harvard Medical School, Charlestown, Massachusetts, USA). The plasmid was transfected with lentivirus packing vectors pCMVdelta8.2 and VSV-G using the Lipofectamine 3000 transfection kit (Invitrogen, USA). Pseudovirus-containing culture medium was collected after 72 h of transfection and the viral titer was estimated. PalMGFP expressing lentivirus was used to generate GFP-MC38 cells. MC38 ($2 \times 10^5$) cells were dispensed into a six-well plate along with an appropriate amount of viral stock in the medium. After selection by puromycin, the cells with the highest expression of GFP were sorted using a BD FACSAria III cell sorter (BD Biosciences, San Jose, CA, USA) and used in this study. GFP expression was further confirmed by confocal fluorescence microscopy (Nikon, Melville, NY, USA). For implanting the MC-38 cell line in mouse colon tissue, 8–12-week-old C57BL/6 male mice ($n = 5$ per group) were anesthetized with a mixture of ketamine and xylazine by intraperitoneal injection and $0.5 \times 10^6$ GFP-MC38 colon cancer cells/ 50 μl in PBS or PBS were administered via endoscopy-guided colonic submucosal injection into the colon via rectum using a 27 G needle[60]. After six weeks, mice were euthanatized and the liver, mesenchymal lymph node, and spleen were collected for histological evaluation.

For AhR (AhRKO) cell, CRISPR/CAS9 plasmids for mouse AhR (sc-419054) were purchased from Santa Cruz Biotechnology Inc. Dallas, TX, USA. Transfection was performed using the plasmids to make AhR knock out cells. AhR knock out was confirmed by Western blot analysis.

**Nanoparticle preparation**. Total lipids from fecal exosomes were extracted with chloroform and dried under vacuum[61]. Totally, 200 nM of lipid was suspended in 200–400 μl of 155 nM NaCl. After ultraviolet (UV) irradiation at 500 mJ/cm$^2$ in a Spectrolinker cross-linker (Spectronic Corp.) and bath sonication (FS60 bath sonicator, Fisher Scientific) for 30 min, the nanoparticles were collected by centrifugation at 100,000g for 1 h at 4 °C.

**Labeling of nanoparticles and fecal exosomes**. Nanoparticle or fecal exosomes were labeled with DIR or PKH26 Fluorescent Cell Linker Kits (Sigma) using the manufacturer's instructions. Nanoparticle or fecal exosomes were suspended in 250 μl of diluent C with 4 μl of DIR or PKH26 dye and subsequently incubated for

30 min at RT[61]. After washing with PBS and centrifugation at 100,000g for 1 h at 4 °C, the pellet was resuspended in PBS and used in experiments.

**In vitro uptake of labeled exosomes by cells**. To study the effect of endocytosis inhibitors[67] (Cytochalasin D (10 μM), Bafilomycin A1 (1 μM), Amiloride (250 μM), and Chlorpromazine (25 μM) on fecal exosomes uptake, primary hepatocytes or Kupffer cells/lines ($2 \times 10^5$) were cultured at 37 °C in the presence of endocytosis inhibitors before the addition of PKH-26 labeled exosomes for ($2 \times 10^6$) an additional 12 h culture period. Cells were washed 3 times with PBS and fixed with 2% paraformaldehyde. Fixed cells were acquired using a BD FACSCanto flow cytometer (BD Biosciences, San Jose, CA) and analyzed using FlowJo software (Tree Star Inc., Ashland, OR).

**Western blot analysis**. The tissues or cells were washed with ice-cold PBS and homogenized. The cells/exosomes were lysed in radioimmunoprecipitation assay lysis buffer with addition of protease inhibitor for 1 h at 4 °C. The crude lysates were centrifuged at 14,000g for 15 min. Protein concentrations were determined using the BioRad Protein Assay Reagent. Samples were diluted in sodium dodecyl sulphate sample buffer. Proteins were separated by 10% sodium dodecyl sulphate-polyacrylamide gel electrophoresis and transferred to nitrocellulose membranes (Bio-Rad). Individual protein was detected with specific antibodies (1:2000 dilution) and visualized by infrared fluorescent secondary antibodies (Supplementary Table 6). The protein bands were visualized and analyzed on an Odyssey CLx Imager (LiCor Inc., Lincoln, NE).

**Flow cytometry**. The liver of mice was perfused with perfusion buffer (($Ca^{2+}$-$Mg^{2+}$ free HBSS containing 0.5 mM EGTA, 10 mM HEPES and 4.2 mM $NaHCO_3$ supplemented with Type I collagenase (0.05%) and trypsin inhibitor (50 μg/ml; pH 7.2)) and then harvested into the complete medium as described elsewhere[60]. Cells isolated from liver tissue were fixed with 2% paraformaldehyde (PFA) and stained with albumin and F4/80 primary antibodies for 40 min at 4 °C. After three washes with PBS, cells were stained with Alexa488 or PE conjugated secondary antibodies for 1 h at RT. Stained liver cells (monocytes and hepatocytes) treated with PKH26$^+$ nanoparticle or fecal exosomes were acquired using a BD FACSCanto flow cytometer (BD Biosciences, San Jose, CA) and analyzed using FlowJo software (Tree Star Inc., Ashland, OR). Uptake of PKH26$^+$ nanoparticles or fecal exosomes was analyzed by using gating strategy shown in Supplementary Fig. 16.

**Confocal microscopy**. For frozen sections, periodate-lysine-paraformaldehyde fixed tissues were dehydrated with 30% sucrose in PBS overnight at 4 °C and embedded into optimal cutting temperature compound. Tissue was subsequently cut into ultrathin slices (5 μm) using a microtome. The tissue sections were blocked with 5% BSA in PBS. Primary antibodies (1:800) were added and incubated at 4 °C overnight. Sections were washed three times followed by secondary antibodies conjugated to a fluorescent dye (at 1:2000 dilution). Nuclei were stained with 4′,6-diamidino-2-phenylindole dihydrochloride (DAPI). For in-vitro cultured cells, $2 \times 10^5$ cells were grown on coverslips in six-well plates and cocultured with PKH26 labeled fecal exosomes for 16 h at 37 °C in a $CO_2$ incubator. Cells were washed with PBS and fixed with 2% PFA. Nuclei were stained with DAPI. Tissues and cells were visualized via confocal laser scanning microscopy (Nikon, Melville, NY).

**Histological analysis**. For hematoxylin and eosin (H&E) staining, tissues were fixed with buffered 10% formalin solution (SF93-20; Fisher Scientific, Fair Lawn, NJ) overnight at 4 °C. Dehydration was achieved by sequential immersion in a graded ethanol series of 70, 80, 95, and 100% ethanol for 40 min each. Tissues were embedded in paraffin and subsequently cut into ultrathin slices (5 μm) using a microtome. Tissue sections were deparaffinized in xylene (Fisher), rehydrated in decreasing concentrations of ethanol in PBS, stained with H&E, and the slides were scanned with an Aperio ScanScope as previously described[60].

**Glucose uptake assay**. Glucose uptake was performed in accordance with the manufacturer's instructions. Glucose Uptake-Glo$^{TM}$ Assay from Promega (J1341) was used. Briefly, $2 \times 10^4$ cells (primary hepatocytes and cell lines) were seeded in complete medium into a 96-well tissue culture plate. Adipocytes and myocytes were differentiated before use for glucose uptake with differentiation medium (DMEM supplemented with 2% horse serum and 10 nm insulin). When cells achieved 50–60% confluency, fecal exosomes ($1 \times 10^6$) or tissues extracts (1 mg/ml) and PBS as control were added and incubated for 16 h at 37 °C in a $CO_2$ incubator. Cells were treated with 1 nM of insulin for an additional 1 h. The medium was removed, and cells were washed twice with PBS. Totally, 50 μl of 2-deoxyglucose (DG, 1 mM per well) was added and incubated for 1 h at RT. Totally, 25 μl of stop buffer was added and mixed briefly, and then 25 μl of neutralization buffer was added and shaken briefly. Totally, 100 μl of 2DG6P detection reagent was added and the mixture was shaken for 3 h at RT. Luminescence was recorded with 135 gain efficiency using a SYNERGY H1 (BioTek) luminometer.

For glucose uptake testing, hepatocytes were cultured with supernatant from monocytes (U937) precultured with fecal exosomes. U937 cells ($2 \times 10^5$) were seeded into six-well plates and when the cells reached 50−60% confluence, fecal

exosomes ($2 \times 10^6$) were added and incubated for 16 h at 37 °C in a $CO_2$ incubator. Culture supernatants were harvested and centrifuged at 100,000$g$ to remove fecal exosomes. These supernates (0.5 ml) were further used in hepatocyte (cultured with $0.5 \times 10^6$ fecal exosomes) cultures to determine which cytokines were induced by fecal exosome treatment of macrophages.

**Cytokine production analysis**. To investigate the effects of fecal exosomes on the regulation of cytokine production in peripheral blood and adipose tissues, peripheral blood was collected from mice orally administered exosomes for 14 days. Plasma and adipose tissues were processed for lysate preparation as per instructions provided in the kit manual. Cytokines were analyzed with a Proteome Profiler Mouse XL Cytokine Array Kit (R&D Systems, ARY028) as per the manufacturer's instructions. Quantification of the spot intensity in the arrays was conducted with background subtraction using HLImage++ (Western Vision Software).

**Macrophage depletion**. For macrophage depletion, Clodrosomes[R] (Encapsula Nano Sciences) was used in accordance with the manufacturer's instructions. In brief, a single intravenous injection (100 µl) of Clodrosomes[R] was given to each mouse and 72 h later macrophage depletion was confirmed by whole blood staining of F4/80 and analysis by flow cytometry.

**Enzyme-linked immunosorbent assay (ELISA)**. Tumor necrosis factor (TNF)-α, interleukin (IL)-6 and IL-10 levels in plasma or tissues were quantified using an ELISA method describe previously[72]. ELISA reagents were purchased from eBioscience and assays were performed in accordance with the manufacturer's instructions. Briefly, a microtiter plate was coated with anti-mouse IL-6, TNF-α and IL-10 antibody at 1:200 overnight at 4 °C. Excess binding sites were blocked with 100 µl/well of blocking solution (PBS containing 0.5% BSA) at RT for 1 h. After washing three times with PBS containing 0.05% Tween 20, sera collected from mice were diluted 2-fold, added in a final volume of 50 µl to the plate wells, and incubated for 1 h at 37 °C. After 3 washes with PBS, the plate was incubated with 100 µl of HRP-conjugated anti-mouse antibody (Pierce) diluted 1:50,000 in blocking solution for 1 h at RT. After the final 3 washes with PBS, the reaction was developed for 15 min, blocked with $H_2SO_4$ and optical densities were recorded at 450 nm using a microtiter plate reader (BioTek Synergy HT).

**Neutralization of TNF-α and IL-6**. The method used for TNF-α and IL-6 neutralization has been described previously[72]. Briefly, to neutralize TNF-α and IL-6 in the conditioned media (CM) before adding the CM to the hepatocytes (FL83B) cultures, the harvested CM was pre-incubated at 37 °C for 1 h with a rat anti-TNF-α antibody (R&D system), a rat anti-IL-6 antibody (R&D System) or with a mixture of both antibodies. The neutralizing dose50 ($ND_{50}$) for the anti-TNF-α antibody was 0.2 µg/ml. For rat anti-IL-6, 1.0 µg ml$^{-1}$ antibody was used based on the neutralizing dose 50 provided by the R&D system. Normal rat IgG at the same concentration as the anti-TNF-α and anti-IL-6 antibodies was used as a control.

**Affymetrix array**. Total mRNA was extracted from tissues using Qiagen total RNA isolation kit (cat. No. 74104). Totally, 100 ng of RNA for each sample submitted to Invitrogen/ThermoFisher Scientific Affymetrix facility, Santa Carla, CA, USA. Transcriptome Analysis Console (TAC) 4.0 from ThermoFisher Scientific was used to analyze the data.

**Surface plasmon resonance (SPR)**. SPR experiments were conducted on an OpenSPR$^{TM}$ (Nicoya, Lifesciences, ON, CA) as previously described[76]. Experiments were performed on a LIP-1 sensor and NTA sensor (Nicoya, Lifesciences). Tests were run at a flow rate of 20 µl/min using HBS running buffer (20 mM HEPES, 150 mM NaCl, pH 7.4). First, the LIP-1 sensor chip was cleaned with octyl β-D-glucopyranoside (40 mM) and CHAPS (20 mM). Liposomes (1 mg/ml) were injected on the sensor chip for 10 min until stable resonance was obtained. After immobilization of nanoparticles (made up of PC and exosomal total lipids), the surface was blocked with BSA (3%) in a running buffer. After a stable signal was obtained, recombinant AhR protein (cat. No. OPCD01209; AVIVA SYSTEMS BIOLOGY, San Diego, CA, USA) was run over the immobilized liposomes. A negative control test was also performed by injecting protein onto a blank sensor chip to check for non-specific binding. After 10 min, the nanoparticle binding proteins were eluted using NaOH (200 µM). For the NTA sensor (protein), AhR protein was injected (0.5 mg/ml) for 10 min until a stable resonance was obtained. After immobilization of protein, nanoparticles were run over the immobilized protein. The sensograms were analyzed using TraceDrawer kinetic analysis software (Nicoya, Lifesciences).

**Direct binding of PC lipid with AhR**. A 5 nM PC or PE lipid was coated onto 96-well plates in 200 µl of 1× coating buffer (cat. No. 00-0044-59, eBioscience) overnight at 4 °C. Wells were washed three times with 1× wash buffer (PBS with 0.05% tween20) and blocked with ELISPOT buffer (cat. No. 00-4952-54) for an hour at RT. After washing the wells, recombinant AhR protein (0.5 mg/ml) in 100 µl of diluent buffer (cat. No. 00-4202-55) was added and incubated for 2 h at RT. After appropriate washing, anti-AhR antibody (1:1000) in 100 µl of diluent buffer was added and incubated for 1 h at RT and subsequently detected with

fluorochrome-conjugated secondary (anti-mouse) antibody, and plates were scanned using an Odyssey Scanner (Licor Bioscience, Lincoln NE).

**Transfer of PC from exosomes to liver and quantification of labeled exosomes in plasma**. Nanoparticles were generated from PC (34:2) and labeled with DIR dye. A standard curve of fluorescence was prepared with known amounts of labeled nanoparticles. In total $2 \times 10^9$ labeled nanoparticles were orally gavaged to each mouse and after 6 h mice were sacrificed and fluorescence was measured in the liver and converted into a number of nanoparticles using a standard curve. In a similar fashion, CD63$^+$A33$^+$ DIR labeled exosomes were used to prepare a standard curve with known numbers of labeled exosomes. In total $2 \times 10^9$ labeled exosomes were orally gavaged to each mice and blood was collected at 0, 0.5, 1, 2, 4, 6, 12, and 24 h and plasma was isolated. Fluorescence in 100 µl of plasma was quantified. This fluorescence was converted to number of exosomes using a standard curve.

  Details of PC (34:2) used in the study
  Hygroscopic: Yes
  Light Sensitive: Yes
  Molecular formula: $C_{42}H_{80}NO_8P$
  Percent composition: C 66.54%, H 10.64%, N 1.85%, O 16.88%, P 4.09
  Purity > 99%
  Stability: 3 months
  Formula weight: 758.060
  Exact mass: 757.562
  Synonyms1-hexadecanoyl-2-(9Z,12Z-octadecadienoyl)-sn-glycero-3-phosphocholine.

**Insulin signaling array**. A 1 µg of total RNA from liver, WAT, and muscle tissues of mice treated with PBS, L-Exo, or H-Exo was reverse transcribed using Superscript III reverse transcriptase (Invitrogen). An insulin signaling array (PAMM030ZE) from Qiagen was performed on an Applied Biosystems ViiA 7 Real-Time PCR System in accordance with manufacturer's instructions.

**Hepatocyte re-expression of AhR by adenovirus vector in $AhR^{-/-}$ mice**. Ad-AhR and Ad-GFP recombinant adenovirus[6] were obtained from the laboratory of Dr. Jongsook Kim Kemper, Department of Molecular and Integrative Physiology, the University of Illinois at Urbana, IL, USA. A single injection of $5 \times 10^9$ plaque-forming units (pfu) of adenovirus carrying AhR (Ad-AhR) or GFP (Ad-GFP) in 200 µl PBS was given to $AhR^{-/-}$ mice via tail vein injection. After 3 days, mice were fed the HFD and oral gavage-given the H-Exo ($2 \times 10^9$/dose/day) for 14 days.

**Lipids analysis in peripheral blood**. Peripheral blood samples of mice were collected into non-heparinized capillary tubes coated with 4% sodium citrate. The levels of cholesterol, triglycerides, ALT and AST were determined by a Piccolo lipid panel plus (Abaxis Inc., USA).

**Statistical analysis**. Statistical significance was determined using the Student's one- or two-tailed $t$ test or one-way analysis of variance with Tukey post hoc test for multiple comparisons. GraphPad Prism 7.0 (GraphPad Software) was used for data analysis. The results were presented as mean ± standard deviation. $P$ values <0.05 were considered statistically significant. * < 0.05, ** < 0.01, *** < 0.001; ****<0.0001. Each data repeated at least three times.

**Reporting summary**. Further information on research design is available in the Nature Research Reporting Summary linked to this article.

## Data availability

All data generated or analyzed during this study are included in this published article (and its supplementary information files or provided in source data file). All data are available from the authors upon reasonable request. Affymetrix array data are available at GSE156848 (https://www.ncbi.nlm.nih.gov/geo/query/acc.cgi?acc=GSE156848). MS proteomics data have been deposited to the ProteomeXchange Consortium via the PRIDE partner repository with the dataset identifier PXD022641. All the lipids identified by the lipidomics analysis shown in Supplementary Data 5. Source data are provided with this paper.

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

## Acknowledgements

We thank Dr. D. Wilkey for mass spectrometry analysis, Dr. Xandra O. Breakefield (Department of Neurology, Massachusetts General Hospital, Charlestown, Massachusetts 02129, USA) for providing lentivirus vectors expressing GFP, Dr. Jongsook Kim Kemper (University of Illinois at Urbana-Champaign, Urbana, IL 61801, USA) for providing pGL3B-PEMT-Luc, and Dr. J. Ainsworth for editorial assistance. We thank Penn Diabetes Research Center (Rodent Metabolic Phenotyping Core, University of Pennsylvania, PA 19104) for the CLAM assay. Funding: This work was supported by a grant from the National Institutes of Health (NIH) (R01AT008617). Huang-Ge Zhang is supported by a Research Career Scientist (RCS) Award and P20GM125504. M.M. and J.M are supported by P50 AA024337 and P20 GM113226. J.P. was supported by the National Institute of General Medical Sciences of the National Institutes of Health (NIH) grant P20GM103436.

## Author contributions

A.K., Y.T., and H.-G.Z. designed the study, analyzed and interpreted the data, and prepared the paper; A.K. and K.S. performed the experiments and interpreted the data; Y.T. analyzed Affymetrix data; M.K.S. performed the GTT and ITT; J.M. provided histological analysis; C.L. and F.X. provided HPLC analysis; M.L.M. performed protein analysis; L.Z., B.W., and X.Q. provided technical support; G.W.D., Y.J., X.Z., C.Q., Z.X., and X.H. provided MS analysis for bioactive lipids. H.B., S.Z., C.M., and J.W.P. interpreted the findings.

## Competing interests

The authors declare no competing interests.
