## [Peer Review File · Nature Communications]

Reviewers' Comments:

Reviewer #1:

Remarks to the Author:

The manuscript "High-fat diet-induced upregulation of exosomal phosphatidylcholine contributes to insulin resistance" by Anil Kumar et al, demonstrated that exosomal lipids play the role in development of insulin resistance. Mechanistically, the authors showed that HFD dramatically changed the lipid profile of intestinal epithelial exosomes from predominantly PE to PC, which resulted in inhibition of the insulin response via binding of exosomal PC to AhR expressed in hepatocytes and suppression of genes essential for activation of the insulin pathway. The study is novel, and the experimental approaches are comprehensive and appropriate. There are a number of significant issues the authors should address to improve the manuscript.

1. It is unclear why fecal exosomes instead of the serum exosomes were focused. If the authors claimed the gut-liver axis, the serum exosomes should be profiled and compared to those of the feces. With the data presented, it is an overstatement that these are intestinal epithelial cell-derived exosomes.
2. In Figure 1, phosphatidylethanolamines (PE) lipids were lower while phosphatidylcholine (PC) lipids were higher in HFD-fed mice compared to those in RCD-fed mice; however, both PE and PC lipids were higher in T2D patients compared to the healthy subjects. Please explain the inconsistency.
3. Although phosphatidylethanolamine N-methyl transferase (PEMT), the transferase enzyme that converts PE to PC, is expressed in the intestine, liver is the among the highest expression tissues. What are the expression and dynamics of PEMT in the liver of the HFD-fed mice? Again, this is important if the authors claim the gut-liver axis effect.
4. In Figure 2, the authors claimed that CD63+A33+ exosomes were preferentially targeting hepatocytes, inhibiting hepatocytes glucose uptake. However, the data showed preferential uptake of the exosomes by Kupffer cells instead of hepatocytes when culturing with H-Exo. Please explain the data.
5. The updates of L-Exo and H-Exo by hepatocytes and macrophages were conducted in cell lines, the experiments should be repeated in primary hepatocytes and macrophages to be more physiologically relevant.
6. The authors showed H-Exo-treated macrophages inhibited hepatocytes glucose uptake, suggesting that the crosstalk between hepatocytes and macrophages may contribute to insulin resistance. Are there any other sources of cytokine production (ex. adipose tissues) which may be possibly affected by H-Exo treatment under obese condition?
7. The authors suggested preferential uptake of H-Exo by macrophages, but the induction of AhR was demonstrated to be induced in the hepatocytes. Are macrophages expressed AhR and is the macrophage expression of AhR induced by H-Exo? Is the AhR induction secondary to the uptake and activation of macrophages by H-Exo?
8. The authors implied that H-Exo mediated activation of AhR leads to inducing dyslipidemia in mice. However, the activation of AhR has been reported to improve insulin sensitivity despite exacerbating hepatic steatosis (Hepatology 2015; 61: 1908-1919). Discussions should be balanced.
9. Please examine expression of AhR downstream target genes (ex. Cyp1a1, Cyp1a2, Cyp1b1) to confirm the activation of AhR by H-Exo.

10. In Figure 5D, the dose-dependent increase of AhR under PC treatment looked not significant by Western blot. Please show quantifications.

11. In Figure 6B, expression of several genes in addition to IRS2 was down-regulated in H-Exo-treated mice. Please explain the rationale of selecting IRS2 as a target of AhR-mediated insulin resistance upon H-Exo treatment. The authors didn't provide enough evidence showing the relationship between IRS2 and AhR. It's a pure speculation and this limitation should be discussed.

Reviewer #2:

Remarks to the Author:

This manuscript is a study that PC in exosomes derived from gut epithelial cells plays a key role in insulin resistance induced by high fat diet (HFD).

Questions:

1. Exosomes are a kind of extracellular vesicles (EVs). Fecal EVs are composed of human cell-derived and microbe-derived EVs. A previous report by Choi et al (Scientific Report 2015) showed that microbial EVs are a key players in the development of insulin resistance.

- Question 1): What is the relationship between epithelial cell-derived exosomes and microbial EVs in the development of insulin resistance induced by HFD?

- Question 2): Authors described the method of microbial EV exclusion. Additional evidence is needed to verify no contamination of microbial EVs in your exosomes. For example, in terms of genes in EVs, exosomes harbor 18S rDNA, but microbial EVs 16S rDNA. So you need to show data about 16S or 18S rDNA.

- Question 3): By quantitative PCR with 16S and 18S rDNA of fecal EVs, you need to show the composition of host cell exosome vs. microbial EVs.

2. There is a controversy whether lipid or sucrose components in a Western diet induce insulin resistance. In your study, authors applied HFD. Is there any sucrose component in your HFD recipe? If HFD contained sucrose, the role of sucrose will be described in the development of insulin resistance.

3. Authors suggested that exosomes released from gut epithelial cells predominantly trafficked to liver and small amount to adipose and muscle tissues. How is the route of gut epithelial cell exosomes from gut to liver, i.e., via lymphatics or portal veins?

4. Authors suggested that PC carried by exosomes is targeted and delivered to specific cells, such as hepatocytes, whereas, the free form of PC is not.

- Question 1): Why do exosomes, but no PC itself, target hepatocytes?

- Question 2): In terms of exosome endocytosis by hepatocytes or macrophages, what is the mechanism of endocytosis? Via receptor-mediated endocytosis or exosomal lipid and host cell lipid interaction?

Reviewer #3:

Remarks to the Author:

The manuscript by Anil Kumar et al. reported one of the potential mechanisms of insulin resistance induced by a high-fat diet. The authors reported that the lipid composition of exosomes from intestinal epithelial cells changes with a high-fat diet predominantly shown by the increase of phosphatidylcholine (PC). These trends were identified in mice with a high-fat diet and humans with type 2 diabetes compared to healthy individuals. Then, the authors examined the role of exosomes from high-fat diet mice in insulin resistance by investigating preferential uptake of PC-enriched vesicles by Kupffer cells, up-regulation of cytokines (primary IL-6 and TNF-alpha), and

AhR pathways. This is a well-organized study that carefully examined possible explanations of why a high-fat diet affects insulin sensitivity. Although there might be many other factors, the change of lipid composition and relevant pathways could shed light on a better understanding of underlying mechanisms.

I have a few comments to improve the understanding of the results.

1. In Figure 1, the authors showed a significant increase of PC in exosomes from intestinal epithelial cells. How about the lipid composition of the originating cells? What could be the underlying mechanism of the increase of PC?
2. In Figure S3, total and CD63+A33+ exosome counts were studied. How were the exosome counts measured? According to Figures S3C-E, it looks like majority exosomes (>80%) are from intestinal epithelial cells defined by CD63+A33+. The positive ratio is surprisingly high in these samples, especially considering not all exosomes may carry CD63. Please provide detailed information about the experiments and discuss the results.
3. In Figure 2, lean mice were given by exosomes from high-fat diet mice at a dose of 2×10^9 vesicles in 200 μ l. How is the dose determined? How many mice were used in each testing group?

Minor comments:

some acronyms need to be defined (e.g., AhR, LPS).

Colors in sensorgrams in Figures 5F and G need to be changed. Almost impossible to differentiate between yellow/brown, purple/pink, etc.

In Fig. 5J, which expression was detected between AhR or phosphorylated-AhR?

Although lyophilic dyes (i.e., DiR or PKH26 used here) are often used to label exosomes, other researchers also report issues with lyophilic dyes, including self-assembly lipophilic dyes. To address the issue, please provide co-location images of dye and CD63 staining in exosomes.

Reviewer #4:

Remarks to the Author:

The manuscript from Kumar et al explores the role of intestinal exosomes in insulin resistance. The authors showed in mice that high-fat diet modified the lipid composition of intestinal exosomes, which are enriched in phosphatidylcholine (PC). PC enrichment is also observed in the stool exosomes of T2D patients compared to healthy subjects. Using multiple mice and cell models the authors showed that only exosome containing PC induced insulin resistance. Exosome from high-fat diet mice are predominantly taken up by liver macrophages and hepatocytes, which leads to the inhibition of insulin signaling pathway in particular by IRS-2. After showing that PC-enriched exosomes bind to the AhR receptor, the authors have shown that this receptor is necessary for the effects of these exosomes on insulin signaling pathway.

The work carried out by Kumar et al is impressive and provides new evidence concerning the mechanisms involved in insulin resistance during obesity. The data presented gave rise to specific comments and several methodological aspects need to be clarified.

Major comments:

-Page 3 and 4: The authors indicated in Results section that exosomes are purified from feces through sucrose gradient however the method is missing in Materials & Methods section.

-Table S1: This table presents a list of proteins, but quantitative and/or qualitative differences in the composition of proteins between L-exo and H-exo are necessary.

-Page 5: "Moreover, mouse colon epithelial (MC-38) cells treated with the fecal metabolites from HFD mice showed increased expression of the luciferase gene, driven by the PEMT promoter (Figure S2K). » What are these fecal metabolites? There is no indication in Materials & Methods

section. What are the links with exosomes?

-Page 5 and table page 37, T2D patients and healthy subjects: There is no information regarding the clinical parameters of the two groups of subjects. The table (numbering is missing) is not informative enough and healthy group is missing. There is no information in the manuscript concerning ethical authorization and consent of participants in compliance with the Declaration of Helsinki principles.

-Page 6 and Materials&Methods: "dose" need to be better explained. If I understand correctly 10 doses= 10 x 0.2ml so a mouse received by gavage 2 ml a day? Is it correct?

-page 6: What are the effects of H-Exo administration to mice submitted to regular (chow) diet ? Is there a change in insulin sensitivity? Are they resistant to insulin?

-Page 8: "Mice were gavage-given human CD63+A33+ T2D-Exo..." These exosomes come from one donor (in this case it is important to indicate which one, and why this donor is chosen) or from a donor pool?

-Page 9: H-Exonano and H-Exonano PC- are obtained after lipid extraction; they are therefore a mixture of lipids and cannot be considered as exosomes. Thus the term "nanoparticules" can no longer be used. The names must be changed to H-Exolipids and H-Exolipids PC-; the term "lipids extracted from H-Exo" can also be used.

Moreover the authors indicated that "a fixed amount of PC was added », but what is this « fixed amount » ?

Table S4 : It is better to use the same names as in the text and therefore replace H-EXOPCDEL and PCB terms. The units are missing. A lipidomic analysis and a quantitative comparison between H-Exolipids (H-Exonano) and H-Exolipids PC- (H-Exonano PC-) would be useful to show all the differences between these two lipid mixtures.

-page 10 : Why mice submitted to GGP-MC38 exosomes into the colon are analyzed 6 weeks post-injection while mice submitted to oral administration of the same exosomes are analyzed after 48h ? Are there differences in the efficiency of two modes of administration? Are the effects observed on the liver after administration into the colon the same as after oral administration?

-page 12 : « ...supplemented with a known amount of synthesized PC (PC+). » What is the nature of this synthesized PC? What is the fatty acid composition of this PC? What is the amount used?

-page 14 :What is the fatty acid composition of PC (34:2) ?

-page 15: What are the differences between PC (34:2) and PC(34:2)nano?

-page 15 and Figure 5H: Is there an interaction between L-Exo and AhR ?

-page 15 and Figure 5A: Is there any induction of well-known AhR target genes ?

-page 16 and 17: What about the phosphorylation of AKT known to be involved in the insulin signaling pathway?

Discussion: PC contains fatty acids and often palmitic acid, which is known to induce insulin resistance. A discussion on the potential involvement of this fatty acid in the effects of PC is necessary. The metabolic effects of H-Exo presented by the authors are obtained after oral gavage, what about the relevance in humans? A discussion of the potential mechanisms involved in the "transfer" of intestinal exosomes from intestinal lumen to circulation would be helpful.

Minor comments

Page3: There seems to be an error in the referencing of the figure, Figure S1 D-L instead of Figure 1D-L ?

Page 3: Table S9 should be indicated in the Results section when High fat diet is mentioned for the first time. What is the composition of RCD diet? References of the diets are required. Ethical authorization must be indicated.

Tables and supplemental tables: In several tables and/or supplemental tables units are missing (for example table S2, S5...)

Reviewer #5:

Remarks to the Author:

The authors describe the contribution of exosomal lipids on the development of insulin resistance in a high-fat diet mouse model. There has been a significant amount of research and statistical investigation performed on behalf of the authors. However, an introduction to the work that has been accomplished and mentioned in other research studies that has influenced this work is not adequately provided.

The authors would greatly benefit from proofreading services as quite a few typos and grammatical errors were found. For example, the authors reference to Figure S1D-L is improperly stated as Figure 1D-L.

The following items need to be appropriately addressed before this manuscript should be published:

- The MS/MS parameters and the following information on the study are missing: [1] type of column used, [2] direct infusion or LC parameters, [3] compound identification software/parameters/workflow, [4] QC procedures, [5] data processing flow, [6] the types of standards used for identification, [7] the types of internal standards used in the study, [8] normalization, etc.
- Every section of this manuscript is missing a more thorough discussion/interpretation of the data/figures/results, their implications, and their connection to previous study results from other authors.
- Figure 1A should be modified to show the percentages of each lipid class. The reader should not have to flip to the supplementation information to obtain this data.
- The mass spectrometer used throughout the manuscript should be referred to as a triple quadrupole.
- In Figure S2, the authors should adopt the LipidMAPS nomenclature of an “_”, which refers to the level of confidence in the lipid annotations based on the type of instrument employed.
- No discussion of how the PE and PC standards were characterized by HPLC is provided. How are the authors sure that the HPLC peak observed for the samples was not contaminated by other an overlap in chromatographic space with other lipid classes? Why wasn't relative quantitation performed using the MS/MS data instead of the HPLC, which has more confidence in lipid class assignments?
- At least one significant digit should be used for the percentages provided in the manuscript. For example, the authors report a PC composition in T2D patient exosomes as 0% when in actuality the supplemental information shows 0.3%.
- How were the differences in the food intake of the mice accounted for in the study in regards to the comparison of the lipid profile?

September 29, 2020

Re. NCOMMS-20-12166

Title: High-fat diet-induced upregulation of exosomal phosphatidylcholine contributes to insulin resistance

Dear Reviewers,

We are grateful to reviewers for the time and effort in reviewing our manuscript. We appreciate your encouraging comments on our research and your valuable suggestions for clarification, correcting typos, and further improvement of our manuscript. Below we provide answers to the comments of each reviewer.

REVIEWER COMMENTS

Reviewer #1 (Remarks to the Author):

The manuscript “High-fat diet-induced upregulation of exosomal phosphatidylcholine contributes to insulin resistance” by Anil Kumar et al, demonstrated that exosomal lipids play a role in development of insulin resistance. Mechanistically, the authors showed that HFD dramatically changed the lipid profile of intestinal epithelial exosomes from predominantly PE to PC, which resulted in inhibition of the insulin response via binding of exosomal PC to AhR expressed in hepatocytes and suppression of genes essential for activation of the insulin pathway. The study is novel, and the experimental approaches are comprehensive and appropriate. There are a number of significant issues the authors should address to improve the manuscript.

1. It is unclear why fecal exosomes instead of the serum exosomes were the focus. If the authors claimed the gut-liver axis, the serum exosomes should be profiled and compared to those of the feces. With the data presented, it is an overstatement that these are intestinal epithelial cell-derived exosomes.

Response: In the revised manuscript, the lipid profile of serum exosomes was compared with the lipid profile of fecal exosomes. In agreement with data generated from fecal exosomes, high-fat diet fed mice also have higher circulating exosomal PC than lean mice (Suppl. Fig. 3b; Suppl. Table 4).

This finding provides a rationale for further investigating whether H-Exo PC circulating exosomes from high-fat diet fed mice may also have a role in the development of insulin resistance. However, the circulating exosomes are a mixed population that could be released from cells that are residents in various tissues including intestinal epithelium, adipose, liver, muscles and many other tissues. Isolating and having enough subsets of cell specific exosomes from circulating and mixed exosomes for this study, in particular

in vivo experiments, is technically challenging. In contrast to the blood exosomes, a large amount of fecal exosomes are released from intestinal epithelial cells which is one of the major subpopulations of exosomes in the gut community that are released daily (1g of feces/20g B6 mouse can be collected per day, <https://www.livescience.com/54909-why-do-mice-poop-so-much.html>, 4×10^{10} CD63⁺A33⁺exosomes/g feces). Additionally, there are relatively much fewer tissues involved in releasing exosomes into the intestinal lumen. These points have been discussed in the discussion section of the revised manuscript, page 28, paragraph 2.

2. In Figure 1, phosphatidylethanolamine (PE) lipids were lower while phosphatidylcholine (PC) lipids were higher in HFD-fed mice compared to those in RCD-fed mice; however, both PE and PC lipids were higher in T2D patients compared to the healthy subjects. Please explain the inconsistency.

Response: In the revised manuscript, we explain this discrepancy in the results section, page 10, paragraph 1. It is known that abnormally high and low cellular PC/PE ratios are linked to disease progression. Although the absolute value of PE from stool exosome samples of T2D patients was increased instead of decreased, when compared to what we observed in the mouse obesity model, the ratios of PC/PE (Suppl. Fig. 4i) in healthy-Exo is lower than that in T2D Exo.

3. Although phosphatidylethanolamine N-methyl transferase (PEMT), the transferase enzyme that converts PE to PC, is expressed in the intestine, liver is among one of the highest expressing tissues. What are the expression and dynamics of PEMT in the liver of the HFD-fed mice? Again, this is important if the authors claim the gut-liver axis effect.

Response: In the revised manuscript, we provide the levels of liver PEMT enzyme quantitatively analyzed by western blot (Suppl. Fig. 3e). The results indicate that PEMT is increased in the liver of mice fed a 60% high-fat diet over 12 months compared with lean mice.

4. In Figure 2, the authors claimed that CD63⁺A33⁺ exosomes were preferentially targeting hepatocytes, inhibiting hepatocytes glucose uptake. However, the data showed preferential uptake of the exosomes by Kupffer cells instead of hepatocytes when culturing with H-Exo. Please explain the data.

Response: We apology for the confusion; the statement has been modified in the revised manuscript. As (Fig. 3b) shows, the H-Exo was taken up by hepatocytes (~38%), as well as by Kupffer cells (>60%). As a consequence of take up, we have shown that take up of H-Exo by Kupffer cells leads to the induction of inflammatory cytokines (TNF- α , and IL-6). The uptake of H-Exo by hepatocytes that leads to induction and activation of AhR signaling. The activation of the AhR signaling pathway leads to inhibition of the genes involved in the insulin signaling pathway by downregulating several genes such as the IRS2, PI3K and AKT mediated pathways.

5. The uptake of L-Exo and H-Exo by hepatocytes and macrophages was conducted in cell lines, the experiments should be repeated in primary hepatocytes and macrophages to be more physiologically relevant.

Response: The results presented in the Fig. 3e & f were, in fact, generated from primary hepatocytes and Kupffer cells. Moreover, Suppl. Figure 8A - C (in initial manuscript submission) was also generated from primary cells (In the revised manuscript, Suppl. Fig. 9a -c). **We have clarified this matter in the revised manuscript.**

6. The authors showed H-Exo-treated macrophages inhibited hepatocyte glucose uptake, suggesting that the crosstalk between hepatocytes and macrophages may contribute to insulin resistance. Are there any other sources of cytokine production (ex. adipose tissues) which may be possibly affected by H-Exo treatment under obese condition?

Response: In the revised manuscript, a cytokine array (Fig. 4a, right panel) of adipose tissue extracts was performed, followed by an ELISA for analysis of TNF- α and IL-6 (Suppl. Fig. 15a). The results indicate that white adipose tissue from H-Exo treated mice has increased levels of the above-mentioned inflammatory cytokines.

7. The authors suggested preferential uptake of H-Exo by macrophages, but the induction of AhR was demonstrated to be induced in the hepatocytes. Are macrophages expressing AhR and is the macrophage expression of AhR induced by H-Exo? Is the AhR induction secondary to the uptake and activation of macrophages by H-Exo?

Response: In the revised manuscript, the expression of AhR in macrophages before and after H-Exo treatment was PCR quantified over time (3h, 6h & 12h). qPCR was performed for AhR, TNF- α and IL-6 expression in liver macrophages. These results suggest that TNF- α and IL-6 expression preceded the induction of AhR (3h versus 12h) upon uptake of H-Exo (Suppl. Fig. 14a-c).

8. The authors implied that H-Exo mediated activation of AhR leads to inducing dyslipidemia in mice. However, the activation of AhR has been reported to improve insulin sensitivity despite exacerbating hepatic steatosis (Hepatology 2015; 61: 1908-1919). Discussions should be balanced.

Response: In the revised manuscript, we discussed the role of AhR signaling in the discussion section, page 27 and paragraph 1. In brief, H-Exo PC induced AhR signaling activation contributes to the development of insulin resistance, but constitutive expression of AhR in AhR transgenic mice has been shown to improve insulin sensitivity despite exacerbating hepatic steatosis by induction of FGF21 (Hepatology 2015; 61: 1908-1919). The aryl hydrocarbon receptor (AhR) is a ligand-activated transcription factor that is activated by small molecules provided by the diet, microorganisms, metabolism, and pollutants. A number of studies have demonstrated that control of AhR functions is challenged by the fact that AhR is often involved in balancing opposing processes. Depending on the nature of factors that activate AhR, different panel of genes with

different functions could be induced upon AhR activation. In addition, negative or positive feedback loops could inhibit/amplify the AhR signal. Therefore, whether Exo PC effects AhR signaling may be altered by different tissue microenvironments needs to be further studied.

9. Please examine expression of AhR downstream target genes (ex. Cyp1a1, Cyp1a2, Cyp1b1) to confirm the activation of AhR by H-Exo.

Response: in the revised manuscript, we provide the qPCR analyzed data for the expression of Cyp1a1, Cyp1a2, and Cyp1b1 (Suppl. Fig. 13b)

10. In Figure 5D, the dose-dependent increase of AhR under PC treatment did not appear significant by western blot. Please show quantifications.

Response: In the revised manuscript, the western blot data (Fig. 5d) was analyzed quantitatively and the ratio to β -actin was added.

11. In Figure 6B, expression of several genes in addition to IRS2 was down-regulated in H-Exo-treated mice. Please explain the rationale of selecting IRS2 as a target of AhR-mediated insulin resistance upon H-Exo treatment. The authors didn't provide enough evidence showing the relationship between IRS2 and AhR. It's a pure speculation and this limitation should be discussed.

Response: The action of insulin is initiated by binding to its cognate receptor. The activated receptor, in turn, recruits and phosphorylates a panel of substrate molecules. Among these, IRS-1 and IRS-2 appear to be the adapter molecules playing a major role in the coupling to the PI3K-PKB downstream kinases. Tyrosine phosphorylated IRS1/2 recruits the heterodimeric p85/p110 PI3K at the plasma membrane, where it produces the lipid second messenger PIP3, that in turn activates the serine/threonine protein kinase B (PKB)/Akt. Akt regulates the insulin-stimulated translocation of the glucose transporter GLUT-4 at the plasma membrane resulting in increased glucose uptake.

In the revised manuscript, our data (Fig. 6c -f; C57BL/6 mice) show that H-Exo treatment led to inhibition of expression of IRS-2 and IRS-2 downstream genes such as PI3K and Akt activation which play an important role in insulin signaling mediated uptake of glucose. These effects including inhibition of expression of IRS-2 were abolished in the liver of AhR^{-/-} mice (Fig. 6c -f; AhR^{-/-} mice), suggesting AhR cross-talk with the insulin signaling pathway. Further, the evidence for AhR involvement in the insulin signaling pathway and that AhR downregulates the IRS-2 expression was derived from experiments where AhR^{-/-} mice had AhR re-expressed via an Ad-AhR tail injection. Western blots of liver extracts from these mice suggest that re-expression of AhR led to a reduction in pIRS-2 protein levels (Fig 6g). In the discussion section (page 22, paragraph 2 & 3), we further discussed the significance of this finding. This finding provides a foundation for further studying whether AhR cross-talk with the insulin signaling pathway is also modulated via H-Exo dependent expression of other molecules such as peroxisome proliferator-activated

receptor-gamma (PPAR γ), srebf1, lep, jun, and ldlr (Fig. 6b). These molecules have been known to regulate insulin signaling too.

Reviewer #2 (Remarks to the Author):

This manuscript is a study that PC in exosomes derived from gut epithelial cells plays a key role in insulin resistance induced by high-fat diet (HFD).

Questions:

1. Exosomes are a kind of extracellular vesicle (EVs). Fecal EVs are composed of human cell-derived and microbe-derived EVs. A previous report by Choi et al (Scientific Report 2015) showed that microbial EVs are key players in the development of insulin resistance.

- Question 1): What is the relationship between epithelial cell-derived exosomes and microbial EVs in the development of insulin resistance induced by HFD?

Response: It is well documented that high-fat diet feeding changes the composition of the gut microbiome. The microbial EVs released from gut bacteria could be taken up by intestinal epithelial cells and subsequently lead to altering the composition of epithelial cell-derived exosomes ¹. In this study, our research focused on the effect of fecal exosomes on the insulin response. Since our results in this study show that gut microbiota seem to not affect the H-Exo mediated insulin response, or the data generated from germ-free C57BL/6 mice versus C57BL/6 mice, we did not further determine whether the role of microbial EVs had an effect in the H-Exo mediated insulin response.

- Question 2): The authors described the method of microbial EV exclusion. Additional evidence is needed to verify no contamination of microbial EVs in their exosome preparations. For example, in terms of genes in EVs, exosomes harbor 18S rDNA, but microbial EVs 16S rDNA. So you need to show data about 16S or 18S rDNA.

Response: In the revised manuscript, additional evidence that support that there is no contamination of microbial EVs in exosomes is provided.

In addition to the markers we provided in our initial submission. i.e., CD63 and A33 markers, we are providing additional evidence that exosomes used in the study were not contaminated with microbial-derived EVs or products.

1. MS/MS analysis indicated that after fecal exosome pull-down with CD63⁺A33⁺ antibody, neither microbial-derived proteins (Suppl. Table 1) nor microbial-derived products were identified when compared with the gut microbial EV protein published profile ^{2,3}
2. qPCR shows that no 16S rDNA was detected in CD63⁺A33⁺ exosomes (Suppl. Fig. 2g).
3. LPS, a microbial-derived component, was not detected in the CD63⁺A33⁺exosomes (Suppl. Fig. 2i).

Question 3): By quantitative PCR with 16S and 18S rDNA of fecal EVs, you need to show the composition of host cell exosome vs. microbial EVs.

Response: In the revised manuscript, 16S and 18S rDNA of fecal EVs including CD63+ and CD63- EVs was quantified by qPCR (Suppl. Fig 2g & h), and the composition of mouse intestinal epithelial cell exosomes was shown using a miRNA array (Suppl. Fig. 2l and Suppl. Table 2).

2. There is a controversy whether lipid or sucrose components in a Western diet induce insulin resistance. In reported study, the authors used a HFD. Is there any sucrose component in your HFD recipe? If HFD contained sucrose, the role of sucrose should be described in the development of insulin resistance.

Response: The high-fat diet we used in this study contained sucrose (Please refer to Suppl. Table 8 that shows the high-fat diet composition). In this study, our research focused on the effect of lipids derived from fecal exosomes on the insulin response. The results generated from this study open new avenues for further identifying the factor(s) from a high-fat diet including sucrose that leads to altering the composition of intestinal epithelial cell derived exosomes. These points have been discussed in the revised manuscript, discussion section, page 25.

3. The authors suggested that exosomes released from gut epithelial cells predominantly trafficked to liver and small amounts to adipose and muscle tissues. How is the route of gut epithelial cell exosomes from gut to liver, i.e., via lymphatics or portal veins?

Response: In the revised manuscript, we did experiments using the protocol we published in 2015 JEV, <https://pubmed.ncbi.nlm.nih.gov/26610593/>. Our data (Suppl. Fig. 8a) show that H-Exo trafficked to liver via the portal vein.

4. The authors suggested that PC carried by exosomes is targeted and delivered to specific cells, such as hepatocytes, whereas, the free form of PC is not.
- Question 1): Why do exosomes, but no PC itself, target hepatocytes?
- Question 2): In terms of exosome endocytosis by hepatocytes or macrophages, what is the mechanism of endocytosis? Via receptor-mediated endocytosis or exosomal lipid and host cell lipid interaction?

Response: In the discussion section of the revised manuscript, page 25, we discussed the difference between exosomal associated PC and free PC. Free lipid in general including PC can move down a concentration gradient into the tissue. The longer the distance from lipids is where there is the highest concentration, the less the biological effect will be. In contrast, PC concentrated in epithelial cell-derived exosomes of mice fed with high-fat diet is predicted to not change until it is taken up by recipient cells such as liver macrophages and hepatocytes. Our data show that ~85% H-Exo gain entry via phagocytosis which was found to be inhibited by Cytochalasin D (Fig. 3d) compared to

other inhibitors. Our results show that PC lipid is a predominant factor for H-Exo being taken up by macrophages. This result also agrees with the result published by other groups⁴⁻⁶. The mechanism underlying exosomal PC mediated phagocytosis needs to be further investigated.

Reviewer #3 (Remarks to the Author):

The manuscript by Anil Kumar et al. reported one of the potential mechanisms of insulin resistance induced by a high-fat diet. The authors reported that the lipid composition of exosomes from intestinal epithelial cells changes with a high-fat diet predominantly shown by the increase of phosphatidylcholine (PC). These trends were identified in mice with a high-fat diet and humans with type 2 diabetes compared to healthy individuals. Then, the authors examined the role of exosomes from high-fat diet mice in insulin resistance by investigating preferential uptake of PC-enriched vesicles by Kupffer cells, up-regulation of cytokines (primary IL-6 and TNF-alpha), and AhR pathways. This is a well-organized study that carefully examined possible explanations of why a high-fat diet affects insulin sensitivity. Although there might be many other factors, the change of lipid composition and relevant pathways could shed light on a better understanding of underlying mechanisms.

I have a few comments to improve the understanding of the results.

1. In Figure 1, the authors showed a significant increase of PC in exosomes from intestinal epithelial cells. How about the lipid composition of the originating cells? What could be the underlying mechanism of the increase of PC?

Response: In the revised manuscript, the lipid profile from A33 positive gut epithelial cells was analyzed and the data is presented in Suppl. Fig.3a and Suppl. Table 4. We show that a high-fat diet induced expression of PEMT is one of the possible mechanisms underlying the increase in PC (Suppl. Fig. 3d). This finding opens a new avenue to identify what factors derived from a healthy and unhealthy diet could regulate the expression of PEMT.

2. In Figure S3, total and CD63+A33+ exosome counts were studied. How were the exosome counts measured? According to Figures S3C-E, it looks like the majority of the exosomes (>80%) are from intestinal epithelial cells defined by CD63+A33+. The positive ratio is surprisingly high in these samples, especially considering not all exosomes may carry CD63. Please provide detailed information about the experiments and discuss the results.

Response: In the revised manuscript, information about the experiments as requested by this reviewer is described in detail. Intestinal epithelial cells predominately release CD63 positive exosomes⁷. The results presented in Suppl. Fig. 2j - k is further discussed in the discussion section, page 28, paragraph 2.

3. In Figure 2, lean mice were given exosomes from high-fat diet fed mice at a dose of 2×10^9 vesicles in 200 μ l. How is the dose determined? How many mice were used in each testing group?

Response: Adult mice (25 g body weight) can defecate 0.25 grams to 1 gram of fecal matter a day. Our data show that 3.5×10^{10} exosomes can be isolated from one g of feces. So, we used the lowest numbers of fecal exosomes which can be defected per day. Each experiment has at least 5 mice/group.

Minor comments:

Some acronyms need to be defined (e.g., AhR, LPS). Colors in sensorgrams in Figures 5F and G need to be changed. Almost impossible to differentiate between yellow/brown, purple/pink, etc.

Response: We appreciate the suggestion and we have defined the acronyms in the manuscript. The quality of Fig. 5f & g was improved as per the reviewer's suggestion.

In Fig. 5J, which expression was detected between AhR or phosphorylated-AhR? Although lyophilic dyes (i.e., DiR or PKH26 used here) are often used to label exosomes, other researchers also report issues with lyophilic dyes, including self-assembly lipophilic dyes. To address the issue, please provide co-location images of dye and CD63 staining in exosomes.

Response: In figure 5J of the revised manuscript, total AhR was analyzed. Also, in the revised manuscript, we did the co-location images of dye and CD63 staining as a control (Suppl. Fig.12a).

Reviewer #4 (Remarks to the Author):

The manuscript from Kumar et al explores the role of intestinal exosomes in insulin resistance. The authors showed in mice that high-fat diet modified the lipid composition of intestinal exosomes, which are enriched in phosphatidylcholine (PC). PC enrichment is also observed in the stool exosomes of T2D patients compared to healthy subjects. Using multiple mice and cell models the authors showed that only exosome containing PC induced insulin resistance. Exosome from high-fat diet mice are predominantly taken up by liver macrophages and hepatocytes, which leads to the inhibition of insulin signaling pathway in particular by IRS-2. After showing that PC-enriched exosomes bind to the AhR receptor, the authors have shown that this receptor is necessary for the effects of these exosomes on insulin signaling pathway. The work carried out by Kumar et al is impressive and provides new evidence concerning the mechanisms involved in insulin resistance during obesity. The data presented gave rise to specific comments and several methodological aspects that need to be clarified.

Major comments:

-Page 3 and 4: The authors indicated in the Results section that exosomes are purified from feces through sucrose gradient however the method is missing in the Materials & Methods section.

Response: We apologize for the missing information. In the revised manuscript, the method is described in the Materials & Methods section, page 4.

-Table S1: This table presents a list of proteins, but quantitative and/or qualitative differences in the composition of proteins between L-exo and H-exo are necessary.

Response: Ratio of L-Exo vs H-Exo protein from MS data is calculated and presented in Suppl. Table 1.

-Page 5: “Moreover, mouse colon epithelial (MC-38) cells treated with the fecal metabolites from HFD mice showed increased expression of the luciferase gene, driven by the PEMT promoter (Figure S2K). » What are these fecal metabolites? There is no indication in the Materials & Methods section. What are the links with exosomes?

Response: In the Materials & Methods section of revised manuscript, the fecal metabolites are referred to as the fecal supernatants containing exosomes after the 6000g centrifugation for 30 min. The detailed protocol for harvesting the fecal metabolites is described in the Materials & Methods section, page 5.

-Page 5 and table page 37, T2D patients and healthy subjects: There is no information regarding the clinical parameters of the two groups of subjects. The table (numbering is missing) is not informative enough and the healthy group is missing. There is no information in the manuscript concerning ethical authorization and consent of participants in compliance with the Declaration of Helsinki principles.

Response: In the revised manuscript, all of this information is provided in the Materials & Methods section, page 4.

-Page 6 and Materials & Methods: “dose” needs to be better explained. If I understand correctly 10 doses= 10 x 0.2ml so a mouse received by gavage 2 ml a day? Is it correct?

Response: Each mouse received orally 0.2ml of H-Exo (2×10^9), one dose per day for 14 days. This procedure is described in detail in the Materials & Methods section page 13 under subheading “adoptive transfer of exosomes”.

-page 6: What are the effects of H-Exo administration to mice submitted to regular (chow) diet? Is there a change in insulin sensitivity? Are they resistant to insulin?

Response: When H-Exo is given to RCD mice they become insulin resistant but need 28 days to show a similar effect as when given to mice fed with the high-fat diet. The results are shown below.

-Page 8: “Mice were gavage-given human CD63+A33+ T2D-Exo...” These exosomes come from one donor (in this case it is important to indicate which one, and why this donor is chosen) or from a donor pool?

Response: It was a pool of exosomes harvested from feces derived from seven T2D patients. This information is provided in the Materials & Methods section, page 14.

-Page 9: H-Exonano and H-Exonano PC- are obtained after lipid extraction; they are therefore a mixture of lipids and cannot be considered as exosomes. Thus the term “nanoparticules” can no longer be used. The names must be changed to H-Exolipids and H-Exolipids PC-; the term “lipids extracted from H-Exo” can also be used. Moreover the authors indicated that “a fixed amount of PC was added », but what is this « fixed amount » ?

Response: The terminology has been changed to H-Exo^{lipids} and H-Exo^{lipids PC-}. Also, “a fixed amount of PC was added” has been changed to “known amount (40 nmols)”.

Table S4: It is better to use the same names as in the text and therefore replace H-EXOPCDEL and PCB terms. The units are missing. A lipidomic analysis and a quantitative comparison between H-Exolipids (H-Exonano) and H-Exolipids PC- (H-Exonano PC-) would be useful to show all the differences between these two lipid mixtures.

Response: The changes have been made according to this reviewer’s suggestion. Comparison of PC is shown in Suppl. Fig. 5p.

-page 10 : Why mice submitted to GGP-MC38 exosomes into the colon are analyzed 6 weeks post-injection while mice submitted to oral administration of the same exosomes

are analyzed after 48h? Are there differences in the efficiency of two modes of administration? Are the effects observed on the liver after administration into the colon the same as after oral administration?

Response: Using the GFP-MC38 model is another and independent approach to show intestinal epithelial cell-derived exosomes can traffick from the intestinal lumen to liver. It took 5-6 weeks for GFP-MC38 to grow in the colon and release detectable GFP+ exosomes. We did not use this tumor model to further address insulin response related questions or whether exosomal trafficking is dependent on the administration route because the exosomes released from tumor may have different functions compared to exosomes from non-tumor intestinal tissue.

-page 12: « ...supplemented with a known amount of synthesized PC (PC+). » What is the nature of this synthesized PC? What is the fatty acid composition of this PC? What is the amount used?

Response: In the revised manuscript, all this information is provided in the Materials & Methods under transfer of PC section, page 25.

-page 14: What is the fatty acid composition of PC (34:2)?

Response: In the revised manuscript, the composition of PC (34:2) is provided in the Materials & Methods under transfer of PC section, page 25.

-page 15: What are the differences between PC (34:2) and PC (34:2) nano?

Response: PC (34:2) is the lipid purchased from Avanti Polar Lipids Inc. PC (34:2) Nano is the nanoparticles generated from the PC (34:2) lipid.

-page 15 and Figure 5H: Is there an interaction between L-Exo and AhR?

Response: In the revised manuscript, the interaction of AhR and L-Exo is analyzed with SPR technology and the results are shown in Suppl. Fig. 11.

-page 15 and Figure 5A: Is there any induction of well-known AhR target genes?

Response: In the revised manuscript, the expression of Cyp1a1, Cyp1a2 and Cyp1b1 was quantitatively analyzed with qPCR and the data are presented in the Suppl. Fig. 13b.

-page 16 and 17: What about the phosphorylation of AKT known to be involved in the insulin signaling pathway?

Response: This is an excellent question. In the revised manuscript, our data show that H-Exo treatment led to inhibition of expression of IRS-2 and IRS-2 downstream genes such as PI3K and Akt activation which play an important role in insulin signaling mediated glucose uptake by cells. These effects including inhibition of expression of IRS-2 and

phosphorylation of Akt (pAkt) were abolished in AhR^{-/-} mice (Fig.6c - g), suggesting AhR cross-talk with the insulin signaling pathway. In the discussion section (page 27), we further discussed the significance of this finding. This finding provides a foundation for further studying whether AhR cross-talk with the insulin signaling pathway is also modulated via H-Exo dependence on the expression of other molecules such as peroxisome proliferator-activated receptor-gamma (PPAR γ), srebf1, lep, jun, and ldlr (Fig. 6b). These molecules have been known to regulate insulin signaling too.

Discussion: PC contains fatty acids and often palmitic acid, which is known to induce insulin resistance. A discussion on the potential involvement of this fatty acid in the effects of PC is necessary. The metabolic effects of H-Exo presented by the authors are obtained after oral gavage, what about the relevance in humans? A discussion of the potential mechanisms involved in the “transfer” of intestinal exosomes from intestinal lumen to circulation would be helpful.

Response: All of the critical points noted by this reviewer have been addressed in the discussion section, page 25, paragraph 2.

Minor comments

Page3: There seems to be an error in the referencing of the figure, Figure S1 D-L instead of Figure 1D-L?

Response: The corrections have been made in the revised manuscript.

Page 3: Table S9 should be indicated in the Results section when high-fat diet is mentioned for the first time. What is the composition of the RCD diet? References of the diets are required. Ethical authorization must be indicated.

Response: The composition of the diet is described in the Suppl. Table 8. In the revised manuscript, all missing information is provided. Ethical authorization is shown on page 3 – 4.

Tables and supplemental tables: In several tables and/or supplemental tables units are missing (for example table S2, S5...)

Response: In the revised manuscript the units are provided.

Reviewer #5 (Remarks to the Author):

The authors describe the contribution of exosomal lipids on the development of insulin resistance in a high-fat diet mouse model. There has been a significant amount of research and statistical investigation performed on behalf of the authors. However, an

introduction to the work that has been accomplished and mentioned in other research studies that has influenced this work is not adequately provided.

Response: In the introduction of the revised manuscript (page 4, paragraph 2), exosomal lipid contributions to the insulin response has been briefly summarized. Due to word limitation, we apologized for not being unable to cite all works done by other investigators.

The authors would greatly benefit from proofreading services as quite a few typos and grammatical errors were found. For example, the authors reference to Figure S1D-L is improperly stated as Figure 1D-L.

Response: The revised manuscript has been edited by “Nature Publisher Service (certificate is attached) and native English speaker, an immunologist, Dr. Jerald Ainsworth.

The following items need to be appropriately addressed before this manuscript should be published:

- The MS/MS parameters and the following information on the study are missing:

[1] type of column used,

[2] direct infusion or LC parameters,

[3] compound identification software/parameters/workflow,

: [4] QC procedures,

[5] data processing flow,

[6] the types of standards used for identification,

[7] the types of internal standards used in the study,

[8] normalization, etc.

Response: Excellent suggestion. We added all these details in the Materials and Methods section, page 9 -11.

- Every section of this manuscript is missing a more thorough discussion/interpretation of the data/figures/results, their implications, and their connection to previous study results from other authors.

Response: In the revised manuscript, wherever it is possible, we added the discussion or interpretation in detail.

- Figure 1A should be modified to show the percentages of each lipid class. The reader should not have to flip to the supplementation information to obtain this data.

Response: The percentages of each lipid class presented in the Fig. 1 are shown in the revised manuscript.

- The mass spectrometer used throughout the manuscript should be referred to as a triple quadrupole.

Response: The terminology for triple quadrupole is added throughout the manuscript.

- In Figure S2, the authors should adopt the LipidMAPS nomenclature of an “_”, which refers to the level of confidence in the lipid annotations based on the type of instrument employed.

Response: The changes have been made as per the reviewer’s suggestions.

- No discussion of how the PE and PC standards were characterized by HPLC is provided. How are the authors sure that the HPLC peak observed for the samples was not contaminated by other an overlap in chromatographic space with other lipid classes? Why wasn’t relative quantitation performed using the MS/MS data instead of the HPLC are presented in the revised manuscript. Which has more confidence in lipid class assignments?

Response: The triple quadrupole MS data are presented in Fig. 1b, d & f to confirm the HPLC data in the revised manuscript. HPLC Standards were described in the Materials and Methods section.

- At least one significant digit should be used for the percentages provided in the manuscript. For example, the authors report a PC composition in T2D patient exosomes as 0% when in actuality the supplemental information shows 0.3%.

Response: We apology for the inconsistency and have added one significant digit for the percentages Fig. 1e.

- How were the differences in the food intake of the mice accounted for in the study in regard to the comparison of the lipid profile?

Response: Diet-derived changes in the gut lead to positive or negative impacts on health depending on the composition of the diet. Ingesting a high-fat enriched diet for long periods will lead to health issues of obesity and chronic inflammation and could result in the development of T2D. In this study, we compared the lipid profiles of mice ingesting a high-fat diet versus a regular-chow diet. Our research focused on the effect of fecal exosomes on the insulin response. The results indicate that a high-fat diet alters the composition of the exosome lipid profile in which PC plays a causative role in developing insulin resistance. The findings from this study will open up a new avenue for further identifying the diet-derived factor(s) that contribute to the regulation of composition and function of intestinal epithelial cell-derived exosomes. These points have been discussed in the revised manuscript, discussion section, page 28.

References

- 1 Deng, Z. *et al.* Enterobacteria-secreted particles induce production of exosome-like S1P-containing particles by intestinal epithelium to drive Th17-mediated tumorigenesis. *Nature communications* **6**, 6956, doi:10.1038/ncomms7956 (2015).
- 2 Liu, J. *et al.* Proteomic characterization of outer membrane vesicles from gut mucosa-derived *fusobacterium nucleatum*. *Journal of proteomics* **195**, 125-137, doi:10.1016/j.jprot.2018.12.029 (2019).
- 3 Zakhazhevskaya, N. B. *et al.* Outer membrane vesicles secreted by pathogenic and nonpathogenic *Bacteroides fragilis* represent different metabolic activities. *Scientific reports* **7**, 5008, doi:10.1038/s41598-017-05264-6 (2017).
- 4 Dodd, C. E., Pyle, C. J., Glowinski, R., Rajaram, M. V. & Schlesinger, L. S. CD36-Mediated Uptake of Surfactant Lipids by Human Macrophages Promotes Intracellular Growth of *Mycobacterium tuberculosis*. *Journal of immunology* **197**, 4727-4735, doi:10.4049/jimmunol.1600856 (2016).
- 5 Yeon, S. H., Yang, G., Lee, H. E. & Lee, J. Y. Oxidized phosphatidylcholine induces the activation of NLRP3 inflammasome in macrophages. *Journal of leukocyte biology* **101**, 205-215, doi:10.1189/jlb.3VMA1215-579RR (2017).
- 6 Matt, U. *et al.* WAVE1 mediates suppression of phagocytosis by phospholipid-derived DAMPs. *The Journal of clinical investigation* **123**, 3014-3024, doi:10.1172/JCI60681 (2013).
- 7 van Niel, G. *et al.* Intestinal epithelial cells secrete exosome-like vesicles. *Gastroenterology* **121**, 337-349, doi:10.1053/gast.2001.26263 (2001).

Reviewers' Comments:

Reviewer #1:

Remarks to the Author:

The authors have adequately addressed my comments and as a result, the manuscript has improved from its previous version.

Reviewer #2:

Remarks to the Author:

The raised questions were well-defensed by the authors.

Reviewer #3:

Remarks to the Author:

The authors have addressed my comments well, and I have no further questions.

Reviewer #4:

Remarks to the Author:

The authors addressed many items of the first version.
However some important points need to be clarified.

- As a general comment, a statistical analysis of data presented in Supplementary Tables 1, 2, 3, 4, 6, 7 is needed to identify the differences between the conditions. Number of samples and p value are needed.
- Supplementary Table 1: the legend is not explicit enough. What does the term "signal" mean?
- The authors added in their response to reviewers a graph showing the effect of H-Exo administration to mice submitted to regular chow diet. This graphic is informative and should be added in the article.
- The reference of commercial PC (34:2) used in the study need to be added.
- SPR method: a description of the experimental conditions used to analyze the interaction between AhR and PC and lipid nanoparticles is missing in Material and Methods section
- The composition of RCD diet is missing.
- A discussion of the potential mechanisms involved in the "transfer" of intestinal exosomes from intestinal lumen to circulation is needed.

Reviewer #5:

Remarks to the Author:

All comments have been appropriately addressed.

Reviewers' Comments

Reviewer # 1 (Remarks to the Author):

The authors have addressed my concerns and as a result, the manuscript has improved from its previous version.

Response: We appreciate reviewer's time and effort to improve our manuscript.

Reviewer # 2 (Remarks to the Author):

The raised questions were well-defensed by the authors.

Response: We appreciate reviewer's time and effort to improve our manuscript.

Reviewer # 3 (Remarks to the Author):

The authors have addressed my comments well, and I have no further questions.

Response: We appreciate reviewer's time and effort to improve our manuscript.

Reviewer # 4 (Remarks to the Author):

The authors addressed many items of the first version. However some important points need to be clarified.

1. As a general comment, a statistical analysis of data presented in Supplementary Tables 1, 2, 3, 4, 6, 7 is needed to identify the differences between the conditions. Number of samples and p value are needed.

Response: The number of samples has been added in the tables. *P* values have been mentioned in the figure itself or the figure legend. The experimental conditions were defined in the method section.

2. Supplementary Table 1: the legend is not explicit enough. What does the term "signal" mean?

Response: We have revised the legend for Supplementary Table 1. The "signal" means the signal detected in the MS/MS analysis regarding protein concentration.

3. The authors added in their response to reviewers a graph showing the effect of H-Exo administration to mice submitted to regular chow diet. This graphic is informative and should be added in the article.

Response: We have added the mentioned figure as Supplementary Figure 17.

4. The reference of commercial PC (34:2) used in the study needs to be added.

Response: We have added the reference in the supplemental references.

5. SPR method: a description of the experimental conditions used to analyze the interaction between AhR and PC and lipid nanoparticles is missing in the Materials and Methods section

Response: We have added the missing information in the methods section.

6. The composition of the RCD diet is missing.

Response: We have added the RCD diet composition in Supplementary Table 8.

7. A discussion of the potential mechanisms involved in the “transfer” of intestinal exosomes from intestinal lumen to the circulation is needed.

Response: In the discussion section of the revised manuscript (page 28 and page 29 paragraph 1), we have discussed the potential mechanisms involved in the “transfer” of intestinal exosomes from the intestinal lumen to the circulation.

Reviewer # 5 (Remarks to the Author):

All comments have been appropriately addressed.

Response: We appreciate reviewer’s time and effort to improve our manuscript.

In summary, we appreciate the reviewers’ and editors’ critical comments and invaluable suggestions. Clearly, their suggestions improve the manuscript and it is our belief that these changes completely address the editorial requests and criticisms of the reviewers. We look forward to the acceptance and publishing our of manuscript in Nature Communications.

Sincerely,

Huang-Ge Zhang, M.D., PhD
Professor of Microbiology & Immunology
CTRB 309
505 S. Hancock Street
James Brown Cancer Center
University of Louisville
Louisville, KY, 40202
Phone-502 -500-3993